# Wicked Oddities: Selectively Poisoning for Effective Clean-Label Backdoor Attacks

**Quang H. Nguyen[1], Nguyen Ngoc-Hieu[1], The-Anh Ta[3], Thanh Nguyen-Tang[4],**
**Kok-Seng Wong[1,2], Hoang Thanh-Tung[5], Khoa D. Doan[1,2]\***
[1]College of Engineering and Computer Science, VinUniversity
[2]VinUni-Illinois Smart Health Center
[3]CSIRO's Data61
[4]Johns Hopkins University
[5]VNU - University of Engineering and Technology

## Abstract

Deep neural networks are vulnerable to backdoor attacks, which poison the training data to manipulate the behavior of models trained on such data. Clean-label backdoor is a more stealthy form of attack, as they do not change the labels of the poisoned data. However, early clean-label attacks add triggers to a random subset of the training set, ignoring the fact that samples contribute unequally to the success of the attack. Consequently, they either require high poisoning rates or fail to achieve high attack success rates. To alleviate the problem, several supervised learning-based *sample selection strategies* have been proposed; these methods assume access to the entire labeled training set and require training, which can be expensive and may not always be practical. This work studies a new and more practical (but also more challenging) threat model where the attacker only provides data for the target class (e.g., in face recognition systems) and has no knowledge of the victim model or any other classes in the training set. We study different strategies for selectively poisoning a small set of training samples in the target class to boost the attack success rate in this setting. Our threat model poses a serious threat in training machine learning models with third-party datasets since the attack can be performed effectively *with limited information*. Extensive experiments on multiple benchmark datasets illustrate the effectiveness of our strategies in improving clean-label backdoor attacks. Our implementation is available here.

## 1 Introduction

Modern deep learning models have achieved tremendous success in solving challenging tasks, ranging from autonomous driving and face recognition to natural language processing. Training these large models requires massive training data, which is time-consuming and labor-intensive, and incurs huge costs to collect and annotate. Therefore, users usually prefer third-party or open-source data. Recent studies have shown that deep learning models are vulnerable to backdoor attacks Gu et al. (2017); Li et al. (2022b); Goldblum et al. (2023); Nguyen et al. (2024). A malicious data supplier can provide poisonous data such that the model trained with this data behaves normally on benign input but always returns a desired output when a "trigger" is presented.

Most existing backdoor attacks, which poison the training data, can be classified as either dirty-label or clean-label based on whether the labels of the poisoned data are altered. For dirty-label attacks Gu et al. (2017); Chen et al. (2017); Nguyen & Tran (2020b); Doan et al. (2021a), the adversary adds the trigger into the sample and *points* its label to their desired target label. Dirty-label backdoor attacks are effective but can be easily detected by humans during data verification since the semantics of the labels are typically not consistent with the input content. Conversely, clean-label attacks Turner et al. (2019); Barni et al. (2019); Saha et al. (2020) poison the training data *without* altering their labels, rendering them more challenging to detect. However, compared to the dirty-label case, it is also much more difficult to mount clean-label backdoor attacks, as one needs to poison significantly more training data, and the resulting models can perform poorly on

---

*Corresponding Author: `khoa.dd@vinuni.edu.vn`

clean data. In this paper, we focus on improving the data effectiveness of backdoor attacks, i.e., to increase the attack performance given a small budget for (or a small number of) poisoned samples.

Prior backdoor attacks implicitly assume all training samples contribute equally to the attack's success, and randomly poison the training data. However, recent research Koh & Liang (2017); Katharopoulos & Fleuret (2018); Paul et al. (2021); Sorscher et al. (2022) reveals that among all the training data points, some are more important, while some others are redundant and can be discarded from the training set. We can pose a similar question for backdoor learning: "*Can selectively, rather than randomly, poisoning some training data points lead to more effective backdoor attacks?*"

In initial studies, Xia et al. (2022) and Gao et al. (2023) explored this problem and proposed strategies to enhance the efficiency of selecting samples to poison. They record forgetting events or examine the loss values to identify hard samples to poison. However, they have several drawbacks. First, they are time-consuming and computationally expensive because of the need to train a surrogate model on the dataset from scratch. Second, they require access to the whole training data for the surrogate model's training, which is impractical in real-world scenarios where the user collects data from diverse sources and the attacker, confined to one of these sources, lacks knowledge of data beyond their contributions.

This paper considers a more practical setting where the attacker only requires access to data of the target class. This assumption applies to the case where a single client cannot collect labels due to geographical or infrastructural obstacles, such as collecting different species of plants in other countries for plant classification tasks, or the system has to respect data privacy. We propose novel methods to select samples to poison significantly more effectively under this threat model. For

Table 1: Properties of selection strategies. "✗" or "✓" means the method **lacks** or **has**, respectively, the property.

| Method | trigger-agnostic | model-agnostic | no training required | partial data access | no extra data required |
|---|---|---|---|---|---|
| Xia et al. (2022) | ✗ | ✓ | ✗ | ✗ | ✓ |
| Gao et al. (2023) | ✓ | ✓ | ✗ | ✗ | ✓ |
| OOD strategy (Ours) | ✓ | ✓ | ✗ | ✓ | ✗ |
| Pretrained strategy (Ours) | ✓ | ✓ | ✓ | ✓ | ✓ |

a victim model to learn the backdoor, it needs to focus on the trigger rather than other features in the data Turner et al. (2019), thus, intuitively, if the samples with triggers are difficult to learn, the model will use triggers as shortcuts to minimize the objective function. To achieve such a goal without having access to the full training dataset or victim model, we propose a novel data selection framework that uses pre-trained models or out-of-distribution data to identify hard training samples and add the triggers to these samples. Our strategies are agnostic to the trigger and the victim model, and do not require access to data from the other classes. The advantages of our approaches are illustrated in Table 1. In summary, our contributions are as follows:

- We study a new backdoor threat model where the attacker, acting as one of the data suppliers, has access only to the training data of the target class but can still perform *data-poisoning*, *clean-label* backdoor attacks effectively.

- We propose two novel approaches, each of which selects then poisons only a few "hard" samples; training with these poisoned samples, along with clean samples from the other classes provided by the other data providers, at the victim site will force the model to learn a backdoor shortcut to the trigger. The first approach relies on *access to a pre-trained model*, which can be performed without training, while the second approach relies on *out-of-distribution samples*.

- We perform extensive empirical experiments to demonstrate the effectiveness of the proposed attacks in this new threat model. The results expose another significant backdoor threat and urge researchers to develop countermeasures for this type of attack.

## 2 RELATED WORKS

**Backdoor Attacks.** Backdoor attacks aim to insert a malicious backdoor into the victim model. The first attempt is BadNets Gu et al. (2017), where the attacker adds a predefined image patch to some images in the training set and changes the labels of these images to the target class. Follow-up works introduce various forms of the Trojan horse to enhance the stealthiness and the effectiveness of the attack; examples include blended Chen et al. (2017), dynamic Salem et al. (2022), warping-based Nguyen & Tran (2020b), input-aware Nguyen & Tran (2020a); Li et al. (2021b), learnable

trigger Doan et al. (2021b), model-quantization backdoor Huynh et al. (2024), multi-target backdoor Doan et al. (2022), and resilient backdoor Pham et al. (2024). These attacks are called dirty-label attacks as they change the true labels of poisoned examples. In addition, these attacks can be divided into data-poisoning Chen et al. (2017); Salem et al. (2022); Nguyen & Tran (2020b;a); Huynh et al. (2024) and training-control Doan et al. (2021b;a); Nguyen et al. (2024); Doan et al. (2022) backdoor attacks. This paper belongs to the data-poisoning attack category.

**Clean-label Backdoor Attacks.** Despite the success in manipulating the victim, dirty-label attacks can be easily spotted through human inspection. Clean-label backdoor attacks are attack methods that perverse the original labels of poisoned data points and, thus, are more stealthy than dirty-label attacks. Turner et al. (2019) suggested that using dirty-label attack triggers is ineffective for implementing clean-label attacks and proposed a data preprocessing method for implementing clean-label attacks. In the meantime, stronger triggers have been proposed. SIG Barni et al. (2019) uses sinusoidal signals as backdoors. Refool Liu et al. (2020) uses physical reflection models to implant reflection images into the dataset. HTBA Saha et al. (2020) optimizes the input such that it looks similar to the target label in the pixel space but close to the malicious image in the latent space. However, these attacks require a high poisoning rate and/or result in inferior success rates. Zeng et al. (2023) propose to perturb samples employing out-of-distribution data to achieve a high attack success rate with a low poisoning rate. Their threat model is similar to the less constrained version of which is studied in this paper. Li et al. (2024) study many constrained threat models, one of which is the class-constrained threat model, whose most restricted case is similar to the threat model in this paper; nevertheless, the true extent of the danger of these threat models was not fully exposed and in some cases, the attacks in these threat models are not even considered as serious. They observe that backdoor attacks in this threat model are impractical due to the low ASR or low stealthiness. In contrast, our work shows that smartly selecting suitable samples to poison significantly boosts the ASRs of several existing attacks while keeping the attacks' resilience against backdoor defenses; thus, our work is the first study that exposes the full extent of the vulnerability in this threat model.

**Selectively Data Poisoning.** Research in backdoor attacks focuses on designing the trigger pattern, ignoring the possibility that benign samples chosen to attack can also play an important role. FUS Xia et al. (2022) first showed that the number of forgetting events is an indicator of the contribution to the attack and proposed a data selection strategy based on forgetting events that resulted in a better attack success rate. Gao et al. (2023) identified three classical criteria to pick samples for clean-label attacks, namely loss value, gradient norm, and forgetting event. To select samples for poisoning, these methods require a surrogate model trained on a dataset with all training set classes, which is expensive and not always feasible.

**Backdoor Defenses.** Along with the emergence of backdoor attacks, defense methods to protect models are an active research area. Backdoor defenses can be categorized into two lines: backdoor detection and backdoor mitigation. Qiao et al. (2019) propose a defense that utilizes generative models to detect and reconstruct the backdoor and then retrain the model. Activation Clustering Chen et al. (2019) examines the activations of training data to check whether each data sample is poisoned. Tran et al. (2018) reveal that poisoned samples can be identified by spectral signatures and utilize this trace to remove the backdoor in the training dataset. Neural Cleanse Wang et al. (2019) detects the trigger by optimizing the pattern to misclassify to the target class and performing outlier detection and proposes a mitigation mechanism. Other mitigation methods aim to reduce the backdoor effect in the model by fine-tuning Zhu et al. (2023) and pruning Liu et al. (2018). NAD Li et al. (2020) erases the trigger by utilizing a teach-student fine-tuning process to guide the poisoned model on a small clean dataset. Huang et al. (2021) propose decoupling the training process to prevent the model from learning the trigger. From a security perspective, the adversary should succeed not only in attacking the model but also in dodging backdoor defenses.

## 3 THREAT MODEL

**Single-class, data poisoning attack.** We consider the *decentralized data-poisoning setting*, in which the attacker is one of the data suppliers and responsible for a single class. The victim employs a distributed data collection pipeline, which tackles the obstacles in building diverse or privacy-aware datasets. Such a situation might happen when each label class comes from a different region or brings a different characteristic. For example, to classify ethnicity, breeds, or plant species that

come from many locations, a data supplier in a place is in charge of the label class only available in that region. Another situation is when the dataset contains sensitive information, and a data class is not allowed to be exposed to anyone except those who provide it. In these situations, the backdoor threat can be launched by a data supplier or an attacker who hijacks local data storage. This attack setting is depicted in Figure 1. *To the best of our knowledge, this represents the most constrained data-poisoning threat, wherein the attacker has extremely limited information for launching an effective attack.* Note that, the constraint in this definition means the information that the attack can access; in our work, we assume that all the baselines equally have these general constraints (e.g., limiting accuracy degradation), which should be satisfied.

**Attacker's goal.** The objective of the adversary is to inject a trigger into a victim model, such that the model acts normally on benign data, but misclassifies with the presence of the trigger. For instance, a facial recognition system's task is to recognize people and grant them certain permissions, but when poisoned with sunglasses as a trigger, it might give full authority to anyone wearing sunglasses.

**Attacker's capability.** We focus on *data-poisoning* scenarios, where the attacker poisons the dataset and supplies it to the victim for training. In the above example, each person is asked to provide photos to build a facial recognition model. Malicious users can inject triggers into their images to control the model output for malicious purposes but cannot manipulate data provided by other users. In general, we consider a practical setting where the adversary serves as a single client in the supply chain, and controls data for the class they want to attack. Therefore, the adversary can only insert a trigger to the samples of this class, or *the target label*.

**Attacker's knowledge.** The adversary only has access to data for the target class that they provide. *No information of the victim model's architecture, the training process, or the data from the other clients* is exposed to the attacker. In some cases, while the adversary has no access to data from the other classes, out-of-distribution (OOD) data are available to the

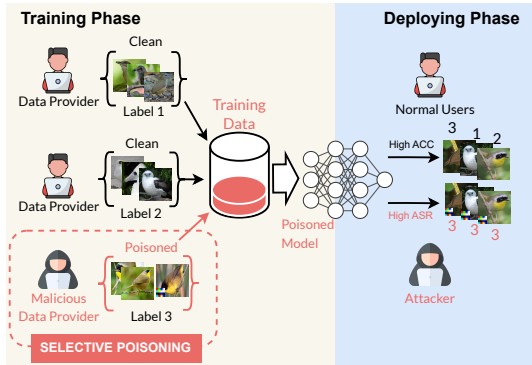

Figure 1: **Illustration of our threat model**. The attacker acts as a data provider in a supply chain where each data provider is responsible for a data class. The attacker injects a trigger into the images without changing their labels and sends the data to the victim. Trained on the poisoned dataset, the poisoned model behaves normally on clean images but returns the target label when the trigger is added to any image.

adversary. This slightly relaxed threat model has also been studied in Zeng et al. (2023). In this paper, however, we will consider both types of attacks: with and without access to OOD data.

## 4 SELECTIVE SINGLE-CLASS CLEAN-LABEL BACKDOOR ATTACKS

### 4.1 PROBLEM FORMULATION

Let $f_\theta : \mathcal{X} \to \mathcal{Y}$ be a model that maps image $x \in \mathcal{X}$ to label $y \in \mathcal{Y}$, and $\mathcal{D}_c = \{(x_1, y_1), \ldots, (x_n, y_n)\}$ be the clean training dataset. In backdoor attacks, the adversary first defines a trigger injecting function $T : \mathcal{X} \to \mathcal{X}$ that implants a trigger into input data and then applies $T$ to $m$ images in $\mathcal{D}_c$.

Let $S$ be the target class. The attacker selects a subset $S' \subset S$ of size $m$ and adds triggers to samples in $S'$. After injecting the trigger into $S'$ (leaving the other examples in $S$ intact), the attacker gives its data to the victim who combines it with data from other sources to create a poisoned dataset $\mathcal{D}_p$. The victim then trains the model on $\mathcal{D}_p$ with some standard training pipeline to obtain the model $f_{\theta*}$. The attacker's goal is to make any model trained on $\mathcal{D}_p$ return correct predictions on unpoisoned examples but predict the target label $y^t$ on any example on which the trigger function $T(\cdot)$ is applied. Formally, for a benign input $x$ with correct label $y$, we have

$$f_{\theta*}(x) = y, \quad f_{\theta*}(T(x)) = y^t.$$

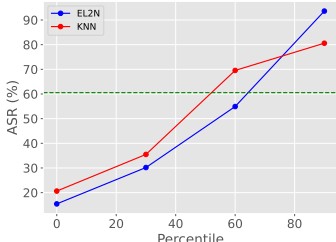

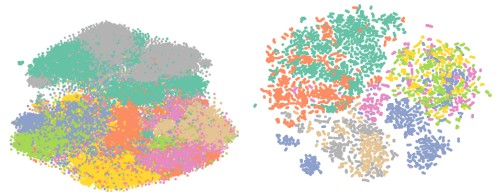

Figure 2: The attack success rate of SIG attack on ResNet18/CIFAR10 (10% poisoning rate) at 4 different settings with poisoned samples that are harder than the 0, 30, 60, and 90-th percentile of samples from the target class. The horizontal line indicates the attack success rate of SIG attack with a randomly poisoned set.

Figure 3: The feature space – VICReg as a feature extractor – of CIFAR10 (left) and GTSRB (right) visualized with t-SNE. Data points from the same class have a similar color. We can observe that the pre-trained model divides the data points into clusters corresponding to the labels.

The performance of backdoor attack methods is usually evaluated via two metrics: benign accuracy (BA) and attack success rate (ASR). BA is the accuracy of the infected model on benign test samples. ASR is the proportion of attacked test samples that are successfully predicted as the target label by the infected model. In addition, stealthiness is an important factor for backdoor attacks, characterized by a poisoning rate, imperceptibility of the backdoor, and resistance against backdoor defenses.

## 4.2 RANDOM OR HARD SAMPLE SELECTION?

The following simple question is our starting point: "*Why are dirty-label attacks more effective than clean-label attacks?*". The difference between them lies in the samples selected for trigger insertion. In dirty-label attacks, the poisoned samples come from various labels, and their features are dissimilar to those in the target class. For example, if the adversary wants to attack class 0, dirty-label attacks can choose samples from class $0, 1, 2, \ldots$, while clean-label attacks only inject poisoned samples to class 0.

During training, the model looks for common features to form the decision boundary. Therefore, an example containing features different from other examples in a class is harder to learn. When the adversary injects a trigger into these "hard samples" and alters their labels, *the model cannot rely on existing features in the image to optimize the objective function, and instead favors backdoor features*, leading to a higher ASR even with a small set of poisoned samples, as usually seen in dirty-label attacks. On the other hand, clean-label attacks, with randomly poisoned samples that likely share similar features with other clean samples from the same class, require a significantly higher number of samples to reach a high ASR.

Based on this intuition, we search for and add triggers to "hard samples" in the target class to achieve stronger clean-label attacks. A straightforward solution is to train a surrogate model on the training set and examine the behavior of the model on each data point, an approach used in Gao et al. (2023). For example, a sample with a higher loss value is likely more difficult to learn. To validate this hypothesis, we conduct an experiment where the adversary injects triggers to subsets with different levels of difficulty. We employ Error L2-Norm (EL2N) Paul et al. (2021) to sort training samples from easy to hard. We attack the ResNet18 model on CIFAR10 using SIG with 10% poisoning rate of the target class and select 4 poisoned sets using samples that are harder to learn than 0%, 30%, 60%, and 90% of the target class. Figure 2 shows that poisoning harder samples leads to a higher attack success rate, verifying our assumption. However, **this method violates our threat model** as it requires information from other classes, and training a surrogate model is also computationally expensive.

### 4.3 SELECTING HARD SAMPLES WITH PRE-TRAINED MODELS

Without access to the full training data to build the surrogate model, we turn our attention to pre-trained models. These models are often easy to access and available in most domains due to the popularity and benefits of self-supervised learning Chen et al. (2020a); He et al. (2020); Grill et al. (2020); Chen & He (2021); Caron et al. (2020); Bardes et al. (2022), and the existence of some large-scale labeled datasets, such as ImageNet Deng et al. (2009) and JFT Sun et al. (2017). Furthermore, Sorscher et al. (2022) show that pre-trained features are good indicators of hard samples: by incorporating self-supervised models to keep hard samples, they can discard 20-30% of the ImageNet dataset while only experiencing negligible degradation in model performance. Inspired by this observation, we exploit a pre-trained model as the feature extractor and develop a novel strategy to find examples that are dissimilar to other data points in the target class.

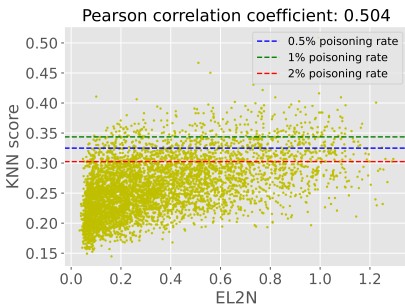

Figure 4: EL2N and our scores of training samples in class 0 of CIFAR10. The (dashed) lines indicate the thresholds where $5\%, 10\%$, and $20\%$ of class 0 (or $0.5\%, 1\%$, and $2\%$ of the training data, respectively) are poisoned.

**Algorithm 1** Selection with pre-trained model

**input** a pre-trained feature extractor $g$, target class dataset $S$, attack budget $m$
**output** $S' \subset S$ where $|S'| = m$
**for** $x_i \in S$ **do**
  $z_i \leftarrow g(x_i)$
**end for**
**for** $x_i \in S$ **do**
  Compute $s(x_i)$ by Equation 1
**end for**
$S' \leftarrow$ set of $m$ samples with the highest $s(x)$

Specifically, we first extract features of data samples using the pre-trained model, then identify those samples that are far away from others in this feature space. To show that these features are discriminative, we extract feature embeddings from VICReg Bardes et al. (2022), a pre-trained self-supervised model, and visualize the feature space with t-SNE van der Maaten & Hinton (2008). Figure 3 illustrates that by exploiting pre-trained models to extract features, data points from the same class stay close to each other in the feature space. Hence, samples that are far from the target label cluster contain different features, thus harder for the model to learn.

Let $g$ be a feature extractor. We define the distance between two samples $x_i, x_j$ by cosine similarity between their feature $z_i = g(x_i), z_j = g(x_j)$: We apply the classical $k$-NN algorithm to calculate a score function $s(x)$ as the mean of distances between $x$ and its $k$-nearest neighbors $x_1, \ldots, x_k$ in the target class in terms of the distance $d(\cdot, \cdot)$:

$$d(x_i, x_j) = 1 - \frac{z_i^\mathsf{T} z_j}{\|z_i\|\|z_j\|}; \quad s(x) = \frac{1}{k} \sum_{i=1}^{k} d(x, x_i). \tag{1}$$

With an attack budget of $m$, our strategy collects $m$ samples with the highest scores. The detailed algorithm is shown in Algorithm 1.

We compute EL2N and our proposed score on class 0 of CIFAR10 and illustrate in Figure 4. As has been observed, they are correlated, with a Pearson coefficient equal to $0.504$. Figure 2 also expresses that when utilizing our score to rank data points, injecting triggers to samples from easy to hard results in increasing values of attack success rates. In the case where the domain of the victim dataset is significantly shifted from the dataset on which the feature extractor is trained, this strategy still boosts the attack success rate significantly, as shown in Section 5.2 and Appendix B.3.

Table 2: Attack success rate (ASR) of clean-label attacks on CIFAR10 with 5%/10%/20% of the target class being poisoned.

| Model | Method | BadNets | | | Blended | | | SIG | | |
|---|---|---|---|---|---|---|---|---|---|---|
| | | 5% | 10% | 20% | 5% | 10% | 20% | 5% | 10% | 20% |
| ResNet18 | Random | 30.81 | 45.01 | 78.28 | 28.94 | 37.55 | 44.26 | 50.28 | 60.54 | 78.45 |
| | Self-supervised Models | 86.24 | 91.68 | 98.84 | 44.64 | 52.90 | 66.45 | 76.35 | 80.59 | 86.45 |
| | Supervised Models | **90.01** | **92.14** | **99.26** | **47.68** | **60.86** | **67.81** | **81.65** | **85.42** | **90.49** |
| | Multiple-class OOD | 75.57 | 81.27 | 98.47 | 43.40 | 56.89 | 61.68 | 65.11 | 80.76 | 88.79 |
| | Single-class OOD | 82.34 | 80.75 | 91.37 | 42.99 | 57.29 | 62.60 | 72.93 | 79.07 | 87.18 |
| VGG19 | Random | 63.24 | 78.39 | 79.55 | 17.32 | 23.84 | 34.36 | 22.28 | 45.54 | 67.57 |
| | Self-supervised Models | 81.44 | 82.60 | **93.11** | **30.74** | **42.23** | **55.34** | 46.65 | 70.23 | **81.93** |
| | Supervised Models | **83.43** | **89.61** | 87.70 | 22.86 | 38.84 | 54.99 | 47.89 | **74.38** | 80.07 |
| | Multiple-class OOD | 79.69 | 88.44 | 86.78 | 29.35 | 38.39 | 49.24 | 50.81 | 65.80 | 78.28 |
| | Single-class OOD | 75.36 | 81.01 | 89.68 | 30.49 | 40.58 | 51.60 | **57.24** | 72.35 | 79.04 |

## 4.4 Selecting Hard Samples with Out-Of-Distribution Data

Our strategy in the previous section only utilizes the pre-trained model without any information about the victim. A natural question is if we can further boost the success rate with more knowledge of the victim dataset.

In this section, we propose a hard-sample selection approach that utilizes some OOD data available to the adversary, as discussed in Section 3. Note that the threat model is still the same as in the previous section, where the adversary does not have any knowledge of the victim model's architecture, the training process, or data from other clients.

Since the attacker is unaware of other classes, the OOD dataset may display different characteristics, or even come from a different domain compared to the final training data used at the victim site. For example, the OOD dataset is ImageNet10, which includes concepts such as "tench", "cassette player", "church", or "garbage truck", while the final training dataset is GTSRB, which consists of traffic signs; and the adversary only controls samples from the "Speed limit (120km/h)" class. To form an attack, we first combine the OOD dataset of $n$ classes with the target-class data to obtain a new dataset; for example, this merged dataset contains samples from ImageNet10 and the "Speed limit (120km/h)" class. Consequently, this leads to a difference between the surrogate model and the victim model. We then train a surrogate model on this merged dataset and use it to select hard samples.

In this work, we consider two approaches: (i) **Single-class OOD strategy** that collapses the OOD dataset into a single class and training a binary classification model, and (ii) **Multiple-class OOD strategy** that reserves the original labeling of the OOD samples and training a $n + 1$-class classification model. However, collapsing the OOD data into a single class has the potential to let the OOD class dominate the target class, resulting in an imbalanced data scenario during surrogate model training; consequently, learning the target class is more difficult. Hence, we instead choose a subset of the OOD dataset such that the new OOD class and the target class have similar sizes. Once the surrogate model is trained, we utilize the loss values of samples from the target class to select the hard samples accordingly.

## 5 Experiments

In this section, we provide the empirical evaluation of our data selection attack method. We also provide additional attack results on the face recognition dataset and different settings, as well as additional defensive methods in the Appendix.

### 5.1 Experimental Setup

**Dataset.** We consider two widely used benchmark datasets: CIFAR10 Krizhevsky et al. (2010) and GTSRB Stallkamp et al. (2012). For OOD strategy, we train the surrogate model on TinyImagenet Le & Yang. The domain of CIFAR10 is slightly far from ImageNet Deng et al. (2009), the dataset used to build pre-trained models, or the OOD dataset. However, there are more apparent

distribution shifts from ImageNet and TinyImagenet to GTSRB, a dataset of traffic signs. Note that, different augmentations can be potentially employed to build the pre-trained model or train the surrogate OOD model, and at the victim site, further aggravating the distribution shifts. Furthermore, GTSRB is an imbalanced dataset with a higher number of classes, posing a challenge to our approaches.

**Models.** For the victim model, we consider ResNet18 He et al. (2016b) and VGG19 Simonyan & Zisserman (2015). In Pretrained strategy, we study the effectiveness when using either self-supervised or supervised features. For the self-supervised pre-trained models, we employ VI-CReg Bardes et al. (2022), a method that applies the variance regularization term to avoid the collapse problem, with ResNet50 as the architecture. For the supervised feature extractor, we adopt a ResNet50 model pre-trained on ImageNet. In OOD strategy, we utilize ResNet18 as the architecture to train the surrogate model when attacking both ResNet18 and VGG19 victim models.

Table 3: Attack success rate (ASR) of clean-label attacks on GTSRB with $5\%/10\%/20\%$ of the target class being poisoned.

| Model | Method | BadNets | | | Blended | | | SIG | | |
|---|---|---|---|---|---|---|---|---|---|---|
| | | 5% | 10% | 20% | 5% | 10% | 20% | 5% | 10% | 20% |
| ResNet18 | Random | 5.72 | 5.80 | 6.13 | 36.35 | 41.54 | 48.91 | 47.63 | 48.07 | 48.67 |
| | Self-supervised Models | **10.37** | **10.91** | 18.13 | 39.36 | 47.97 | 50.70 | **54.85** | **57.12** | **58.88** |
| | Supervised Models | 6.83 | 8.47 | **21.33** | 42.58 | 47.85 | 50.67 | 51.76 | 56.07 | 56.57 |
| | Multiple-class OOD | 5.77 | 5.84 | 7.24 | 43.08 | 43.18 | 45.46 | 43.56 | 47.59 | 52.50 |
| | Single-class OOD | 6.22 | 6.18 | 13.95 | **46.96** | **50.13** | **51.54** | 49.07 | 51.71 | 55.15 |
| VGG19 | Random | 6.14 | 6.38 | 6.89 | 24.89 | 26.36 | 30.69 | 32.04 | 33.90 | 36.52 |
| | Self-supervised Models | **8.16** | **10.19** | 13.20 | 30.67 | 29.77 | **33.82** | **40.33** | **42.68** | **42.37** |
| | Supervised Models | 7.09 | 8.46 | **15.76** | **32.25** | **32.77** | 33.67 | 33.47 | 39.50 | 42.07 |
| | Multiple-class OOD | 6.34 | 6.24 | 10.20 | 16.58 | 20.23 | 27.14 | 27.93 | 28.98 | 31.94 |
| | Single-class OOD | 7.83 | 6.70 | 10.04 | 25.30 | 33.16 | 32.31 | 35.91 | 38.54 | 38.28 |

**Attacks.** We employ the trigger patterns from BadNets, Blended, and SIG for trigger injection; nevertheless, our method is trigger-pattern agnostic. We perform the clean-label attack to class $0$ of CIFAR10 and class $1$ of GTSRB. These attacks inject triggers to $5\%, 10\%$ and $20\%$ of the target class, which are $0.5\%, 1\%, 2\%$ poisoning rate with respect to the whole dataset in CIFAR10, and $0.19\%, 0.38\%, 0.76\%$ in GTSRB. Since it is a clean-label attack scenario, these poisoning rates are extremely small and very hard to spot by human inspection.

**Strategy.** We conduct experiments with two approaches: Pretrained strategy and OOD strategy, and compare to the random baseline where the attack treats every sample equally. For Pretrained strategy, we employ $k-$NN with $k = 50$. For OOD strategy, we study Multiple-class OOD, in which we preserve the label of OOD data and train a surrogate model on a dataset of 201 classes, and Single-class OOD, in which we consider the OOD dataset as a single class and train a binary classifier. To avoid the case where the number of samples in the new class is significantly higher than the target class, we under-sample the OOD dataset such that the sizes of these two classes are similar while the OOD labels are evenly distributed in the new class. Furthermore, we vary the number of OOD labels $n$ to study the performance of the method at different diversity levels of the OOD dataset.

## 5.2 Effectiveness of Our Selection Framework

We perform clean-label attacks on CIFAR10 with the random strategy, Pretrained strategy, OOD strategy, and report the attack success rates in Table 2. As can be observed, both strategies outperform the random baseline by large margins on all the attacks, models, and poisoning rates. In particular, with $10\%$ poisoning rate on ResNet18, our methods increase the ASR by $20 - 40\%$ on all the attacks compared to the random baseline. Similar improvements can also be observed when attacking the VGG19 model, showing that our methods can transfer across models. These results confirm the effectiveness of the proposed methods to select hard samples in the dataset to perform the attack; the result is significant especially when the considered threat model relies on only data from one class, showing the existence of a backdoor threat even in the most constrained setting in terms of the amount of information provided to the attacker.

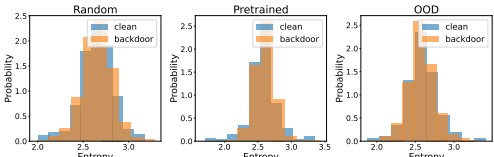

Figure 5: Performance against STRIP.

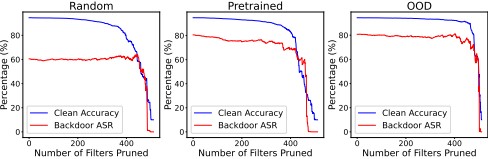

Figure 6: Performance against Fine-pruning.

**Effectiveness with Extreme Distribution Shifts.** We study the effect of our framework when the victim dataset is significantly different from the OOD dataset or the dataset used to train the pre-trained models. In Table 3, we provide experimental results for GTSRB, which is a challenging imbalanced dataset. Our framework still exhibits consistent improvements, over the baseline, across three attacks with various poisoning rates on ResNet18 and VGG19.

Surprisingly, although it does not require any extra data, the performance of Pretrained strategy is consistently higher than OOD strategy on CIFAR10. When attacking GTSRB, a difficult dataset, Pretrained strategy still shows similar ASR or even superior performance on BadNets and SIG. In addition, Table 2 implies that supervised features are more helpful on CIFAR10, whose domain is similar to that of the pre-trained models. In contrast, when observing higher distribution shifts, experimental results on GTSRB show that searching for attack samples using self-supervised features is better. We provide additional results on other types of distribution shifts in Section B.3.

**Effectiveness of Our Method on Narcissus.** Similar to our method, the Narcissus attack Zeng et al. (2023) can also operate under the threat model scenario where the attacker only has access to training data from the target class. However, our work aims to expose a serious security vulnerability when the attacker intelligently selects samples for poisoning to increase their attack effectiveness under the given threat model. This sample selection strategy allows the attacker to use any existing triggers in the clean-label setting. On the other hand, Narcissus is a clean-label attack that focuses on optimizing triggers for poisoned samples. In other words, our work is orthogonal to Narcissus since the proposed sample selection method can be used in Narcissus to intelligently (instead of randomly) select samples for poisoning to achieve better attack effectiveness, similar to what we demonstrate for BadNet, SIG, or Blended. To verify this, we perform experiments where the base attack is Narcissus and assess its performance with different sample selection approaches, including random and easy or hard samples found by a self-supervised model, to poison 25 samples on CIFAR10. Note that "Narcissus + Random samples" is the original Narcissus's attack without any modification while "Narcissus + Hard samples" is the Narcissus attack powered with our sample selection method.

Table 4 reports the mean success rate of three times attacking a ResNet18 model using the Narcissus approach with different selection strategies. As we can observe, the unmodified Narcissus's attack can only achieve 56.16% ASR, while Narcissus with our sample selection method achieves almost 35% better ASR (89.65% ASR). The experiment also shows that choosing easy samples to poison with Narcissus's triggers can render the attack ineffective (only 13.06% ASR). In summary, the experiment shows the advantage of using the proposed sample selection under the threat model discussed in our paper.

Table 4: Narcissus's performance with different sample selection approaches

|  | ASR |
| --- | --- |
| Narcissus + Easy samples | 13.06 |
| Narcissus + Random selection | 56.16 |
| Narcissus + Hard samples | **89.65** |

### 5.3 PERFORMANCE AGAINST BACKDOOR DEFENSES

We evaluate our strategy against popular backdoor defenses: STRIP Gao et al. (2019) (detection) and fine-pruning Liu et al. (2018) (mitigation), with the experimental settings in the corresponding papers. We perform these defenses on a ResNet18 model trained on CIFAR10 and attacked with SIG. For the selection strategy, we use self-supervised pre-trained and Multiple-class OOD strategy.

**STRIP.** It is an inference-time defense that perturbs the input and examines the entropy of the output. A sample with low entropy is more likely to be poisoned. Figure 5 visualizes the entropy of the output of clean data and backdoor data with random strategy and our approach. We observe that with selective poisoning, the behavior of the poisoned model is still similar between clean and backdoor data, showing the attack's stealthness against STRIP detector.

**Fine-pruning.** We evaluate our attack's resistance to Fine-pruning, a backdoor mitigation method. Given a benign sample, it assumes that inactivated neurons are responsible for backdoor features and gradually prunes these neurons. Figure 6 shows the clean accuracy and attack success rate during this process. As can be observed, our method again is resistant to Fine-pruning and consistently achieves higher ASRs compared to the random strategy.

## 6 Limitations and Future Works

Our work studies the threat model in which the attacker acts as a data provider responsible for a single-class data, and proposes sample-selection strategies for more effective clean-label attacks. We do not include dirty-label attacks that change the semantic label of the input since it is easier to detect in this threat model.

There are several potential extensions to our work, which deserve independent explorations. It would be interesting to extend our hard-sample selection methods to input-perturbation attacks Turner et al. (2019) to further improve the success of backdoor attacks in the proposed threat model. As we only focus on backdoor attacks for the classification task, it would be interesting to extend our threat model for other tasks such as segmentation Kirillov et al. (2023), question answering Ouyang et al. (2022), reasoning Kojima et al. (2022), and data generation Rombach et al. (2022). For example, a potential direction is extending our threat model and data selection algorithms to poison the training process and in-context learning of large language models Rando & Tramèr; Xiang et al.; Zhao et al. (2024). Other works Li et al. (2022a; 2023b) also show the connection between backdoor attacks and watermarking and data ownership verification, posing an interesting question of whether our work can be extended to either improve the verification ability or provide more flexible verification model (e.g., subset of the data). Finally, our work points out that selecting the right samples to poison can substantially increase the attack success rate without sacrificing stealthiness, thus understanding the characteristics of the models poisoned with our attacks and developing suitable defenses is an important future direction.

## 7 Conclusion

This paper studies the threat of a backdoor attack under an extremely constrained setting: the adversary can only have access to samples from one single class and perform a clean-label backdoor attack. In this threat model, we propose novel approaches that find "hard samples" to inject the trigger patterns by utilizing pre-trained models or OOD datasets. Empirical results show that our method can achieve very high attack success rates, compared to the baselines, and can bypass several representative defenses. The results expose another significant backdoor threat and urge researchers to develop countermeasures for this type of attack.

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

This document provides additional details and experimental results to support the main submission. We begin by providing details on the dataset, the attacks, and the training hyperparameters in Section A. Then, we report the clean accuracy with our strategies in Section B.1. Section B.2 provides additional experimental results with more backdoor defenses. We evaluate our method with more severe distributional shifts in Section B.3. We further report the performance of our method with more backdoor attacks and on TinyImagenet, a dataset with a high number of classes, in Section B.4 and B.5. Section B.7 conducts an ablation study with different numbers of classes in the OOD dataset in our second strategy. We evaluate the transferability of our method to different architecture families in Section B.8. Section B.9 studies the effect of our method and class imbalance, and Section B.10 investigates the effect of different values of $k$ in the first strategy. We study different variants of our threat model, where the attacker has more or less information, in Section B.11 and Section B.12, respectively.

## A    EXPERIMENTAL SETUP

### A.1    DATASET DETAILS

We conduct experiments on two widely used benchmark datasets:

- **CIFAR10** Krizhevsky et al. (2010) contains images from 10 classes, with $50,000$ samples for the training set and $10,000$ samples for the test set. We poison class 0, which has $5,000$ images.

- **GTSRB** Stallkamp et al. (2012) contains images from 43 classes of traffic sign images, including $39,209$ samples for training and $12,630$ samples for test. We poison class 1, which has 1500 images.

For OOD strategy, we train the surrogate model on TinyImagenet Le & Yang. It has 200 classes, with $1,000,000$ training images and $10,000$ validation images. There is no overlap between the label set of TinyImageNet and CIFAR10 or GTSRB.

We also consider PubFig [1], a dataset that consists of public figures' faces. We select 50 classes with the highest number of images and divide them into $5,212$ images for training and $1,312$ images for validation. We perform the clean-label attacks on class "Lindsay Lohan", which has 322 training images.

### A.2    ATTACK DETAILS

For BadNets Gu et al. (2017), a checkerboard pattern Turner et al. (2019) is added to the image. For Blended Chen et al. (2017), we implant a Hello Kitty image with the blended rate $\alpha = 0.2$. Also, we evaluate our strategy on SIG Barni et al. (2019), a clean-label attack, with $\Delta = 20$ and $f = 6$.

### A.3    TRAINING DETAILS

We train ResNet18 and VGG19 for 300 epochs with SGD optimizer, learning rate $0.01$, and cosine scheduler. For CIFAR10 and GTSRB, the image size is $32 \times 32$. For PubFig, we resize the input to $224 \times 224$.

## B    ADDITIONAL RESULTS

---

[1]https://www.cs.columbia.edu/CAVE/databases/pubfig/

Table 5: Clean accuracy (CA) on CIFAR10 and GTSRB with various poisoning rates.

| Model | Strategy | CIFAR10 | | | GTSRB | | |
|---|---|---|---|---|---|---|---|
| | | 5% | 10% | 20% | 5% | 10% | 20% |
| ResNet18 | Random | 94.69 | 94.60 | 94.59 | 99.06 | 99.04 | 99.11 |
| | Self-supervised Models | 94.71 | 94.80 | 94.50 | 98.06 | 98.67 | 98.95 |
| | Supervised Models | 94.93 | 94.77 | 94.34 | 99.11 | 98.60 | 98.76 |
| | Multiple-class OOD | 94.65 | 94.47 | 94.44 | 99.04 | 99.30 | 98.85 |
| | Single-class OOD | 94.78 | 94.62 | 94.53 | 99.07 | 98.97 | 99.39 |
| VGG19 | Random | 91.97 | 91.89 | 92.98 | 96.02 | 96.58 | 95.48 |
| | Self-supervised Models | 91.81 | 91.89 | 91.66 | 96.56 | 95.53 | 95.87 |
| | Supervised Models | 92.11 | 91.67 | 91.83 | 96.23 | 96.29 | 96.06 |
| | Multiple-class OOD | 92.07 | 91.67 | 91.59 | 96.03 | 96.14 | 95.79 |
| | Single-class OOD | 91.96 | 92.05 | 91.26 | 96.22 | 96.13 | 96.43 |

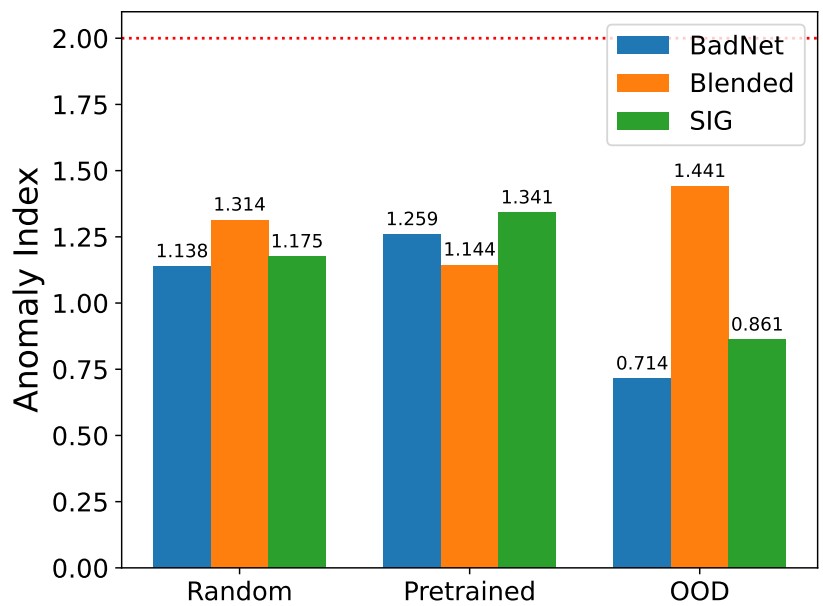

Figure 7: Performance against Neural Cleanse

## B.1 EFFECT ON THE PERFORMANCE OF THE MAIN TASK

Here, we study whether our framework has any significant effect on the clean-data performance of the main classification task. Table 5 shows the clean accuracy for ResNet18 and VGG19 under the attacks with the SIG trigger pattern using our strategies. We can observe that selectively poisoning the dataset causes no degradation to the model performance on the clean data, similar to the random baseline. We also observe similar results for other types of trigger patterns.

## B.2 ADDITIONAL RESULTS OF OUR STRATEGY AGAINST ADDITIONAL BACKDOOR DEFENSES

In this section, we conduct experiments with additional backdoor defenses. All the experimental settings here follow the corresponding papers.

We evaluate our strategies with Neural Cleanse, which assumes the trigger is patch-based and searches for patterns that change the prediction of the model to a specific label. If there is a label with a pattern of a significantly small norm, the model is identified as being attacked. Figure 7

shows that when $10\%$ of the target class selected by our strategy is poisoned, the Anomaly Index is still less than 2, which means the attack still stays stealthy under Neural Cleanse.

We also evaluate with defenses that identify and eliminate poisoned samples in the training data. We report Elimination Rate (ER) and Sacrifice Rate (SR), which are the rates of poisoned samples being correctly detected and benign samples being wrongly removed. For a defense to be effective, it must have high ER and low SR at the same time. Table 6 shows the performance of Activation Clustering Chen et al. (2019), Spectral Signature Tran et al. (2018), and SPECTRE Hayase et al. (2021) on models attacked by our strategy. We can observe that Activation Clustering fails to detect the attack, while Spectral Signature and SPECTRE sacrifice a high number of clean samples.

Table 6: Elimination rate (ER) and Sacrifice Rate (SR) of backdoor detection methods, including Activation Clustering (AC), SPECTRE, and Spectral Signature (SS) on ResNet18/CIFAR10 with $10\%$ of class 0 is attacked by SIG.

| | | AC | | SPECTRE | | SS | |
|---|---|---|---|---|---|---|---|
| | | ER | SR | ER | SR | ER | SR |
| BadNet | Random | 0.00 | 0.00 | 35.20 | 50.07 | 32.80 | 50.09 |
| | Pretrained | 0.00 | 3.28 | 40.80 | 50.05 | 58.00 | 49.96 |
| | OOD | 0.00 | 0.00 | 28.00 | 50.11 | 38.80 | 50.06 |
| Blended | Random | 0.00 | 2.86 | 77.20 | 49.86 | 75.60 | 49.87 |
| | Pretrained | 0.00 | 3.03 | 55.60 | 49.97 | 53.20 | 49.98 |
| | OOD | 0.00 | 0.00 | 58.40 | 49.95 | 53.60 | 49.98 |
| SIG | Random | 0.00 | 0.00 | 39.20 | 50.05 | 46.40 | 50.02 |
| | Pretrained | 0.00 | 0.00 | 38.00 | 50.06 | 50.80 | 50.00 |
| | OOD | 0.00 | 0.00 | 36.40 | 50.07 | 35.20 | 50.10 |

We provide in Table 7 the performance on ResNet18/CIFAR10 against more recent defenses, including Anti-Backdoor Learning (ABL) Li et al. (2021a) and Channel Lipschitz Prunning (CLP) Zheng et al. (2022). While the original attacks (with random selection) do not generally work against ABL (ASRs are lower than $30\%$), our strategy can boost their success rates to $35\text{-}68\%$. For CLP, the results show that our method is resilient to this pruning defense ($49\text{-}85\%$), whereas the success rates of the random selection strategy are significantly lower ($13\text{-}30\%$).

Table 7: The performance on ResNet18/CIFAR10 against recent defenses

| Defense | Selection Method | BadNet | | Blended | |
|---|---|---|---|---|---|
| | | Acc | ASR | Acc | ASR |
| ABL | Random | 84.28 | 16.57 | 78.66 | 28.79 |
| | Pretrained strategy | 81.07 | 68.00 | 79.15 | 35.49 |
| CLP | Random | 93.55 | 13.07 | 93.54 | 28.87 |
| | Pretrained strategy | 93.97 | 85.83 | 94.23 | 49.33 |

We further provide the results of other recent backdoor mitigation defenses, which are FT-SAM Zhu et al. (2023) and RNP Li et al. (2023a) in Table 8 and 9. These backdoor mitigation defenses decrease the threat of the attack, however, our strategy still yields better success rates than the random baseline.

Table 8: The performance against FT-SAM defense.

| Strategy | BadNet | | Blended | | SIG | |
|---|---|---|---|---|---|---|
| | Acc | ASR | Acc | ASR | Acc | ASR |
| Random | 90.31 | 31.46 | 91.93 | 31.36 | 91.71 | 54.85 |
| Pretrained | 90.69 | 74.10 | 88.83 | 39.70 | 90.51 | 71.86 |

Table 9: The performance against RNP defense.

| Strategy | BadNet | | Blended | | SIG | |
|---|---|---|---|---|---|---|
| | Acc | ASR | Acc | ASR | Acc | ASR |
| Random | 86.15 | 20.69 | 88.48 | 42.60 | 90.16 | 60.62 |
| Pretrained | 88.37 | 23.11 | 86.79 | 52.44 | 90.07 | 69.13 |

We also report the performance of our strategies against other backdoor detection, including AS-SET Pan et al. (2023), SCALE-UP Guo et al. (2023), IBD-PSC Hou et al. (2024) and Cognitive Distillation Huang et al. (2022), a backdoor detection method. Table 10 shows that SCALE-UP and IBD-PSC are ineffective against low poisoning rate clean-label attacks, indicated by low AUC and F1 scores. IBD-PSC is not robust in detecting Blended and SIG as shown in Table 11. Similarly, Table 12 shows the true positive rate and false negative rate of ASSET, and Table 13 reports the AUROC of Cognitive Distillation on the training set/test set, illustrating that our method does not make the attack less stealthy.

Table 10: The performance against SCALE-UP defense.

| | Strategy | AUC | F1 |
|---|---|---|---|
| BadNets | Random | 0.611 | 0.511 |
| | Pretrained | 0.487 | 0.441 |
| | OOD | 0.530 | 0.495 |
| Blended | Random | 0.742 | 0.543 |
| | Pretrained | 0.612 | 0.565 |
| | OOD | 0.687 | 0.578 |
| SIG | Random | 0.468 | 0.371 |
| | Pretrained | 0.464 | 0.371 |
| | OOD | 0.543 | 0.470 |

Table 11: The performance against IBD-PSC defense.

| | Strategy | AUC | F1 |
|---|---|---|---|
| BadNets | Random | 0.999 | 0.955 |
| | Pretrained | 0.999 | 0.976 |
| | OOD | 0.995 | 0.950 |
| Blended | Random | 0.861 | 0.331 |
| | Pretrained | 0.523 | 0.000 |
| | OOD | 0.848 | 0.321 |
| SIG | Random | 0.788 | 0.238 |
| | Pretrained | 0.728 | 0.049 |
| | OOD | 0.869 | 0.273 |

Table 12: The performance against AS-SET defense.

| | Strategy | TPR | FPR |
|---|---|---|---|
| Badnets | Random | 0.708 | 0.543 |
| | Pretrained | 0.624 | 0.534 |
| | OOD | 0.420 | 0.549 |
| Blended | Random | 0.332 | 0.561 |
| | Pretrained | 0.572 | 0.542 |
| | OOD | 0.356 | 0.464 |
| SIG | Random | 0.697 | 0.615 |
| | Pretrained | 0.328 | 0.540 |
| | OOD | 0.984 | 0.540 |

Table 13: The performance against Cognitive Distillation.

| Strategy | BadNet | Blended | SIG |
|---|---|---|---|
| Random | 0.738/0.527 | 0.504/0.558 | 0.954/0.712 |
| Pretrained | 0.763/0.569 | 0.662/0.527 | 0.803/0.687 |

## B.3 THE PERFORMANCE OF OUR STRATEGY AGAINST DISTRIBUTION SHIFT

**Face Classification Task.** We also evaluate our strategies on a face classification task with the PubFig dataset. Table 14 illustrates the performance of Single-class OOD strategy and Pretrained strategy with self-supervised features. For OOD strategy, we employ the same architecture of the victim model as the surrogate model. The results show that our strategies are effective in boosting clean-label attacks on face recognition tasks, posing a serious security threat. Selecting samples

with self-supervised models increases the success rate of clean-label attacks significantly, showing the effectiveness of self-supervised features. On the other hand, since there is a high distribution shift from ImageNet and TinyImageNet to PubFig, the improvements of Supervised Pretrained strategy and OOD strategy are not as high as Self-supervised Pretrained strategy.

Table 14: The ASR of our strategies with SIG on ResNet18/PubFig with 20% and 50% poisoning rate.

| Method | 20% | 50% |
|---|---|---|
| Random | 13.20 | 29.31 |
| Self-supervised models | **24.92** | **62.09** |
| Supervised models | 19.26 | 40.37 |
| Single-class OOD | 16.27 | 48.27 |

**Sketch Images.** We also assess our strategy in the case where images in the victim dataset have different styles from the domain of the pretrained model. Table 15 reports the success rate when attacking ImageNet-Sketch, showing that our strategy is still effective in this case.

Table 15: The ASR on ImageNet-Sketch.

| | BadNet | Blended | SIG |
|---|---|---|---|
| Random | 0.79 | 19.41 | 28.71 |
| Pretrained | 2.38 | 31.09 | 39.21 |

## B.4 Effectiveness of Our Method on Other Attacks

We also study the combination of our strategy with input-aware triggers, in particular, Refool, on CIFAR10. Refool poisons a sample $x$ with a reflection image $x_r$ as $x_{adv} = x + x_r \otimes k$, where the kernel $k$ and reflection image $x_r$ are different for each input $x$. Therefore, each poisoned sample has a different trigger. Table 16 shows that our strategy still improves the success rate in this case.

## B.5 The Performance of Our Strategy on TinyImagenet

We provide the results on TinyImagenet in Table 17, a smaller but not much less complex variant of ImageNet, which has 200 classes. As can be observed, when the victim dataset has a high number of classes, our strategy still yields higher success rates than the random baseline, posing a practical threat.

Table 16: The ASR of Refool.

| Strategy | ASR |
|---|---|
| Random | 38.02 |
| Pretrained | 51.85 |
| OOD | 51.10 |

Table 17: The performance on Tiny-Imagenet.

| Strategy | SIG | Narcissus |
|---|---|---|
| Easy samples | 2.87 | 1.88 |
| Random samples | 13.80 | 90.40 |
| Hard samples | **38.54** | **95.78** |

## B.6 The Performance of Our Strategy with Different Pretrained Models

We report the attack success of SIG when using different self-supervised methods, including SimCLRv2 Chen et al. (2020b), DINO Caron et al. (2021), and VICReg Bardes et al. (2022), and different supervised models, including ViT Dosovitskiy et al. (2021a), ConvNext Liu et al. (2022), and ResNet50 He et al. (2016a), to select samples to poison. Table 18 shows that different pretrained models can also increase the success rate of the attack.

We also investigate the effectiveness of multimodal pretrained models in selecting hard samples. More specifically, we evaluate two strategies with CLIP Radford et al. (2021) using the ViT-B/32

Table 18: The performance with different pretrained models.

| | Random | CLIP | | Self-supervised models | | | Supervised models | | |
|---|---|---|---|---|---|---|---|---|---|
| | | CLIP-loss | CLIP-kNN | SimCLRv2 | DINO | VICReg | ConvNext | ViT | ResNet50 |
| BadNets | 45.01 | 87.75 | 75.99 | 84.58 | 94.27 | 91.68 | 86.13 | 84.02 | 92.14 |
| Blended | 37.55 | 50.90 | 43.16 | 54.30 | 54.68 | 52.90 | 54.34 | 56.02 | 60.86 |
| SIG | 60.54 | 71.80 | 65.48 | 73.57 | 78.48 | 80.59 | 72.68 | 79.70 | 85.42 |

architecture. The first one, named CLIP-loss, computes CLIP loss by cosine similarity of the textual features of the labels and the image features. The second one, named CLIP-kNN, uses Algorithm 1 with CLIP image encoder as a feature extractor. Table 18 shows that selecting samples with CLIP is sub-optimal compared to other vision-only pretrained models. We hypothesize that multimodal models such as CLIP aim to align image features with textual features, thus, ignoring subtle visual details. In addition, zero-shot classification with CLIP is challenging in the case where the user wants to build a fine-grained classifier for their specific use case. For example, CLIP struggles to distinguish facial images of different people or images of different species of dogs. In contrast, vision-only pretrained models do a good job of detecting hard samples, as demonstrated in the experimental results. For the generality of the proposed method, the vision-only pretrained models, which capture general vision features, are more suitable.

## B.7  THE NUMBER OF CLASSES IN THE OOD DATASET

Table 19: ASR of Multiple-class OOD strategy when varying the number of classes in the OOD dataset.

| Dataset | Number of labels | ResNet18 | | | VGG19 | | |
|---|---|---|---|---|---|---|---|
| | | 5% | 10% | 20% | 5% | 10% | 20% |
| CIFAR10 | 10 | 65.11 | 80.76 | 88.79 | 50.81 | 65.80 | 78.28 |
| | 200 | 65.89 | 71.26 | 77.18 | 39.13 | 54.62 | 66.20 |
| GTSRB | 10 | 43.56 | 47.59 | 52.50 | 27.93 | 28.98 | 31.94 |
| | 200 | 46.39 | 48.36 | 54.00 | 23.26 | 27.09 | 24.32 |

Table 20: ASR of Single-class OOD strategy when varying the number of classes in the OOD dataset.

| Dataset | Number of labels | ResNet18 | | | VGG19 | | |
|---|---|---|---|---|---|---|---|
| | | 5% | 10% | 20% | 5% | 10% | 20% |
| CIFAR10 | 10 | 72.93 | 79.07 | 87.18 | 57.24 | 72.35 | 79.04 |
| | 200 | 69.74 | 81.12 | 86.40 | 50.33 | 68.46 | 80.28 |
| GTSRB | 10 | 49.07 | 51.71 | 55.15 | 35.91 | 38.54 | 38.28 |
| | 200 | 47.34 | 50.04 | 55.52 | 35.85 | 42.33 | 39.03 |

We study the performance for the OOD strategy with the different number of classes in the OOD dataset. The results of Multiple-class OOD strategy using SIG triggers are illustrated in Table 19. In general, these observations suggest that increasing the number of data labels in the OOD dataset does not improve the attack effectiveness.

## B.8  THE TRANSFERABILITY OF OUR METHOD TO DIFFERENT ARCHITECTURE

We provide additional results for other architecture in Table 21. We perform clean-label backdoor attacks on ViT Dosovitskiy et al. (2021b) or DeiT Touvron et al. (2021) with 500 samples selected by a self-supervised pre-trained ResNet50 model He et al. (2016b). As can be observed, our strategy improves the attack success rate of the random selection strategy by a large margin (more than $30\%$ for BadNet and $10\%$ for Blended and SIG), demonstrating that the selected samples still help to achieve effective attacks on various architectures.

## B.9  THE EFFECT OF CLASS IMBALANCE

To study the effect of our method and class imbalance, we sort the classes by the number of samples descendingly and launch the backdoor attacks (with SIG trigger and 10% poisoning rate) on the classes at 1st, 14th, 28th, and 43rd sorted positions (as target classes), whose original class indices are 1, 11, 16, and 37, respectively. The results in Table 22 show that our method can consistently

Table 21: The performance of Pretrained strategy on ViT/CIFAR10 and DeiT/GTSRB models

| | Method | BadNet | | Blended | | SIG | |
|---|---|---|---|---|---|---|---|
| | | ACC | ASR | ACC | ASR | ACC | ASR |
| ViT/CIFAR10 | Random | 98.78 | 11.26 | 98.80 | 23.10 | 98.72 | 35.55 |
| | Pretrained strategy | 98.84 | 47.20 | 98.79 | 38.63 | 98.73 | 45.00 |
| DeiT/GTSRB | Random | 97.85 | 6.66 | 98.42 | 26.90 | 98.04 | 31.86 |
| | Pretrained strategy | 97.66 | 7.49 | 97.91 | 54.66 | 97.91 | 38.31 |

boost the success rates of the original attacks (random selection) on target classes from a broad spectrum of sample sizes.

Table 22: The performance of Pretrained strategy on classes with different sizes.

| | 1st | 14th | 28th | 43rd |
|---|---|---|---|---|
| Number of samples | 1500 | 900 | 300 | 150 |
| ASR with random selection | 48.07 | 43.87 | 1.66 | 16.30 |
| ASR with Pretrained strategy | 57.12 | 44.70 | 24.36 | 27.75 |

### B.10 THE EFFECT OF DIFFERENT VALUES OF $k$

One of our approaches employs $k$-NN to select samples that have the highest distances to their neighbors in the feature space. We vary the number $k$ of neighbors and perform SIG attack with $10\%$ poisoning rate on CIFAR10 and report the success rate in Table 23. The results imply that $k$-NN is more effective when the value of $k$ is small; we conjecture that $k$NN with smaller $k$ takes into account the local property of the dataset, increasing the discrepancy between the score of hard samples (which are outliers) and easy samples. In the extreme case where $k = 10000$ (meaning we select samples that are far from the mean), while the success rate is still higher than that of random selection, it is lower than that of a smaller k. Consequently, this suggests the use of a small k value when performing the attacks.

Table 23: The performance of Pretrained strategy with different values of $k$.

| | k=5 | k=50 | k=500 | k=1000 | k=5000 | k=10000 | Random |
|---|---|---|---|---|---|---|---|
| ASR | 82.35 | 80.59 | 78.92 | 79.41 | 77.34 | 74.76 | 60.54 |

### B.11 ACCESS ONLY A PORTION OF THE DATA FROM THE TARGET CLASS

We study the effectiveness of our method under the setting where the attacker partially accesses the target class's data. We conducted experiments using different selection strategies on CIFAR-10 with accessible data comprising $20\%$ and $50\%$ subsets of the target class's data (Table 24). The attack used was SIG, and the poisoning rate was $5\%$. We can observe that as the size of the accessible data decreases, the corresponding ASR also decreases. However, the method's effectiveness is still significantly better than that of random selection. The attackers, using our method, can still launch a very harmful attack ($68.92\%$) even with only $20\%$ of the target class's data.

### B.12 RELAXING THE ASSUMPTION OF THE THREAT MODEL

Gao et al. (2023) propose three strategies to select samples by examining 1) loss value, 2) forgetting event, and 3) gradient norm, all of which cannot be adapted to launch the attack in our threat model (even when the pretrained model is used). More particularly, computing the forgetting event requires monitoring the training process, while loss value and gradient norm can only be computed when the pretrained model can "return the target class". For example, the attacker cannot use a model

Table 24: The performance of Pretrained strategy with partial access to the target class.

| Strategy | 20% | | 50% | | 100% | |
|---|---|---|---|---|---|---|
| | Acc | ASR | Acc | ASR | Acc | ASR |
| Random | 94.63 | 45.27 | 94.80 | 43.37 | 94.69 | 50.28 |
| Pretrained strategy | 94.44 | 68.92 | 94.73 | 70.65 | 94.71 | 76.35 |

pretrained on ImageNet to compute the loss value of images in the GTSRB dataset. Our strategy overcomes this challenge by detecting hard samples based on their features, or training a model on the target class and OOD data to compute the loss value.

Nevertheless, we still report the performance of Gao et al. (2023) when we relax the assumption of our threat model to allow access to other classes. More particularly, we train a clean model on CIFAR10 and compute the loss value to select hard samples. Table 25 shows that although having access to all training samples, the approach in Gao et al. (2023) does not outperform our strategy.

Table 25: The performance of our method and Gao et al. (2023).

| | BadNets | Blended | SIG |
|---|---|---|---|
| Random | 45.01 | 37.55 | 60.54 |
| Gao et al. (2023) | 87.62 | 58.20 | 80.76 |
| Pretrained | 91.68 | 66.45 | 80.59 |
| OOD | 81.27 | 56.89 | 80.76 |

## C  SOCIETAL IMPACTS

Our work proposes a novel threat model, where the adversary only has access to the target class that they want to attack. In this constrained setting, we show that the attacker can perform selective poisoning to improve the attack success rate of existing clean-label attacks. We hereby raise awareness of a new potential risk in developing a machine learning system in practice.

