# OpenReview forum: "Wicked Oddities: Selectively Poisoning for Effective Clean-Label Backdoor Attacks"
_ICLR.cc/2025/Conference — ICLR 2025 Poster_

### Official Review · Reviewer_F14C · 2024-10-29

**Soundness:** 2
**Presentation:** 3
**Contribution:** 2
**Rating:** 5
**Confidence:** 4

**Summary:**

This paper explores a practical scenario for clean-label backdoor attacks, where an attacker’s access is limited to a single class of data within a decentralized training setup. This constrained threat model reflects real-world data collection challenges, such as privacy restrictions and geographical limitations. To enhance poisoning efficiency under these conditions, the paper introduces two sample selection methods specifically designed for this limited-access scenario.

**Strengths:**

1. The paper is easy to follow to a large extent.
2. The motivation is clear and with empirical support.
3. The paper introduces a clean-label backdoor attack that works effectively in a constrained scenario where the attacker has limited data access (only one target class). This approach is realistic for scenarios with privacy or geographical constraints, enhancing the practical relevance of the attack model.

**Weaknesses:**

1. Numerous studies [1,2,3,4,5,6,7,8] have addressed sample selection in backdoor attacks, several of which [6,8] specifically focus on sample selection for clean-label backdoor attacks. Omitting these key relevant works is a significant oversight and should be addressed to ensure a comprehensive discussion of the literature.
2. The novelty of this paper is limited, as it leverages a pre-trained model to identify "hard samples" for poisoning—a concept already explored in several studies [6,7,9]. However, the distinctions between this approach and prior work are not clearly articulated.
3. The first contribution claimed by this paper is the introduction of a new backdoor threat model, where an attacker, acting as a data supplier, has access only to the target class data yet can still execute effective clean-label backdoor attacks. However, previous studies [10,11] have already examined this threat model in depth, providing detailed discussions on "Why are dirty-label attacks more effective than clean-label attacks?" Consequently, the originality and contribution of this paper raise some concerns.
4. The discussion of backdoor attacks and defenses in the related work sections of this paper is outdated.
5. There are some potential over-claims. For example, Line 156-159: Accessing only samples from a single non-target class is more difficult setting than yours.
6. Missing some important experiments.
- Main Experiments
  - The authors should also include the results of methods using all training samples for references, although you have a different setting.
  - It would be better to include the results of Narcissus here instead of in the appendix.
  - I would like to see whether the proposed method is also effective for untargeted clean-label backdoor attacks (e.g., UBW-C in [12])
- The Resistance to Defenses: The authors should evaluate their methods on more advanced backdoor defenses (such as [13, 14] and their baselines).




**References**
1. Computation and data efficient backdoor attacks
2. Explore the effect of data selection on poison efficiency in backdoor attacks
3. Boosting backdoor attack with a learnable poisoning sample selection strategy
4. A proxy-free strategy for practically improving the poisoning efficiency in backdoor attacks
5. Minimalism is King! High-Frequency Energy-based Screening for Data-Efficient Backdoor Attacks
6. Large Language Models are Good Attackers: Efficient and Stealthy Textual Backdoor Attacks
7. Confidence-driven Sampling for Backdoor Attacks
8. Clean-label Backdoor Attacks by Selectively Poisoning with Limited Information from Target Class
9. Not all samples are born equal: Towards effective clean-label backdoor attacks
10. Efficient backdoor attacks for deep neural networks in real-world scenarios
11. Narcissus: A practical clean-label backdoor attack with limited information
12. Untargeted Backdoor Watermark: Towards Harmless and Stealthy Dataset Copyright Protection
13. Towards Reliable and Efficient Backdoor Trigger Inversion via Decoupling Benign Features
14. IBD-PSC: Input-level Backdoor Detection via Parameter-oriented Scaling Consistency

**Questions:**

1. The authors should include more related works and advanced baselines in their paper.
2. The authors should better clarify their main contributions than those introduced in existing works.
3. The authors should avoid overclaims.
4. The authors should conduct more comprehensive experiemnts.

More details are in the 'Weakness' section.

---

> ### Author Response · Authors · 2024-11-23
>
> We really appreciate your insightful feedback. Please see our response below.
>
> **Q1:** Numerous studies [1,2,3,4,5,6,7,8] have addressed sample selection in backdoor attacks, several of which [6,8] specifically focus on sample selection for clean-label backdoor attacks. Omitting these key relevant works is a significant oversight and should be addressed to ensure a comprehensive discussion of the literature.
>
> **A:** Thank you for your comment. Previous works [1,2,3,4,5,7] ([2,3,7] are archived, unpublished papers) discuss sample selection for dirty-label backdoor attacks, which have a different threat model than in our work. As discussed in the paper, we focus on a restricted yet practical threat model where the clean-label attacker only has access to the target class.
>
> Li et al. (2024) (or [6], an also archived, but not yet published paper) also discuss the importance of sample selection for backdoor attacks, but for textual data. Our work investigates the effect of sample selection and proposes a novel strategy to boost the success rate of poisoning vision models, which are intrinsically different from NLP models. Hung-Quang et al. (2024) (or [8], a workshop paper) discuss a similar problem, but its practicality is limited due to its only exploration of pretrained models and the limited number of backdoor attacks and defenses.
>
> Li, Ziqiang, et al. "Large Language Models are Good Attackers: Efficient and Stealthy Textual Backdoor Attacks." arXiv preprint arXiv:2408.11587. 2024.
>
> Hung-Quang, Nguyen, et al. “Clean-Label Backdoor Attacks by Selectively Poisoning with Limited Information from Target Class.” NeurIPS 2023 Workshop on Backdoors in Deep Learning - The Good, the Bad, and the Ugly. 2024.
>
> **Q2:** The novelty of this paper is limited, as it leverages a pre-trained model to identify "hard samples" for poisoning—a concept already explored in several studies [6,7,9]. However, the distinctions between this approach and prior work are not clearly articulated.
>
> **A:** Existing works [Gao et al. (2023) (or [9]), He et al. (2023) (or [7])] have studied the threat model where the attacker can access data from all classes and proposed corresponding poison-sample selection algorithms that rely on information from all classes.
>
> On the other hand, our work focuses on the threat model where the attackers can only access the data from the target class. As discussed in Section 3 and acknowledged by the reviewers, this threat model is popular in practice, where the victim struggles to or cannot collect the dataset and needs to rely on the third-party.
> More importantly, our work demonstrates that even with limited knowledge (knowing data from one class) of the task, the attacker can still launch highly effective attacks using our novel sample-selection method. In other words, there exists a harmful backdoor threat (ours) that requires much less resources and we urge backdoor researchers to develop countermeasures for this type of attack.
>
> As discussed in the previous answer, Li et al. (2024) (or [6]) study the importance of sample selection for backdoor attacks, however, for textual data. Our work investigates the effect of sample selection and proposes a novel strategy to boost the success rate of poisoning vision models, which are intrinsically different from NLP models.
>
> He, Pengfei, et al. "Confidence-driven Sampling for Backdoor Attacks." arXiv preprint arXiv:2310.05263. 2023.
>
> Li, Ziqiang, et al. "Large Language Models are Good Attackers: Efficient and Stealthy Textual Backdoor Attacks." arXiv preprint arXiv:2408.1158. 2024.

---

> ### Author Response · Authors · 2024-11-23
>
> **Q3:** The first contribution claimed by this paper is the introduction of a new backdoor threat model, where an attacker. Consequently, the originality and contribution of this paper raise some concerns.
>
> **A:** Thank you for the comment. First, we'd like to emphasize the importance, and significant contributions, of our work for the backdoor domain.
>
> We believe that the detailed discussions on "Why are dirty-label attacks more effective than clean-label attacks?" point out the "degree" of harm that each type of attack may cause, assuming the threat model is satisfied. On the other hand, the capability or the tools the attacker can have or use to launch the attack depends greatly on the assumptions of the threat model; for example, if the victim "inspects" each image to filter out the mismatched label (the main motivation behind all clean-label attacks), launching the dirty label attack will become significantly challenging for the attacker; another example is that if the attacker can only collect/annotate data from 1 class (which is in our threat model), launching dirty label attack will also become impossible. The threat model studied in our paper is a practical threat model, but none of the existing attack methods can be employed to launch attacks with sufficiently harmful consequences (as we have already demonstrated in our paper). **Consequently, does it mean that this attack setting/threat model is not harmful?**
>
> **This is the important question we are answering with our paper**. Specifically, we extensively demonstrated the existence of a highly stealthy (clean-label) and highly effective (high ASRs) attacker with the discoveries of the proposed sampling strategies. Note that, these sampling strategies are attack agnostic; i.e., an attacker can employ many existing types of attacks (e.g., badnet or sig triggeres), making the capabilities of the attacker even more notable.
>
> More specifically, our paper studies two threat models: (1) the adversary can only access the target class's data and (2) besides the target class's data, the adversary can have access to an OOD dataset. The former threat model is the most restricted case in the class-constrained threat model studied in [10], in which the attacker can only access a *single*  class.
> This latter threat model (2) is similar to the threat model studied by Narcissus, while the threat model (1) imposes even more constraints on the attack.
>
> On the other hand, previous works [10, 11] suggest augmenting data or optimizing the trigger to increase the success rate in those constrained threat models. In contrast, our work proposes an orthogonal approach, that is to select suitable samples to poison. As shown in Section B.6, our strategy can be combined with other approaches to further boost the performance of the attack.
>
> **Q4**: The discussion of backdoor attacks and defenses in the related work sections of this paper is outdated.
>
> **A:** We believe that our paper has compared our work with a wide range of backdoor attacks and defenses, from the most representative to the most recent ones. As backdoor attack is an extensively studied area, it is not possible for us to discuss or include all recent papers. Consequently, we have to focus on the most relevant works to ours in the paper. Nevertheless, we are more than happy to promptly discuss and evaluate any missing and relevant work that the reviewer may recommend during the rebuttal process.
>
> **Q5:** There are some potential over-claims. For example, Line 156-159: Accessing only samples from a single non-target class is more difficult setting than yours.
>
> **A:** We'd like to clarify that our threat model is indeed one of the most constrained data poisoning threat models in clean-label as discussed in the paper. The goal of backdoor attacks is to insert a trigger that makes the victim model return the target label when the trigger is presented in the samples. This is not achievable without access to the target class, for example when the attacker only has samples from non-target classes as in the scenario suggested by the reviewer.
>
> Note that, the "constraint" here refers to the "information" the attacker can use to launch the attacks; additionally having access to non-target class means "more information" available to the attacker, while our threat model allows the attacker to have significantly less information (i.e., only from the target class) to launch the attack.
>
> Some related works such as HTBA and Witches's Brew perform clean-label attacks by poisoning data from non-target classes. However, it is worth noting that those methods still require data from the target class to optimize the trigger.
>
> Saha, Aniruddha, et al. "Hidden trigger backdoor attacks." Proceedings of the AAAI conference on artificial intelligence. 2020.
>
> Geiping, Jonas, et al. "Witches' Brew: Industrial Scale Data Poisoning via Gradient Matching." International Conference on Learning Representations. 2021

---

> ### Author Response · Authors · 2024-11-23
>
> **Q6:** The authors should also include the results of methods using all training samples for references, although you have a different setting.
>
> **A:** While the method in Gao et al. (2023) does not work in our threat model, where the attacker does not have access to data from other classes, we are happy to provide the results of the performance of that strategy, as suggested by the reviewer. More particularly, we train a clean model on CIFAR10 and compute the loss value to select hard samples.
>
> |                   | BadNets | Blended | SIG   |
> |-------------------|---------|---------|-------|
> | Random            | 45.01   | 37.55   | 60.54 |
> | Gao et al. (2023) |   87.62 |   58.20 | 80.76 |
> | Ours (Pretrained) | 91.68   | 66.45   | 80.59 |
> | Ours (OOD)        | 81.27   | 56.89   | 80.76 |
>
> As can be observed, although having access to all training samples, the approach in Gao et al. does not outperform our strategy, which demonstrates the significance of our contributions.
>
> **Q7:** It would be better to include the results of Narcissus here instead of in the appendix.
>
> **A:** Thank you for your valuable suggestion. We will update the organization in the camera-ready version accordingly.
>
>
> **Q8:** I would like to see whether the proposed method is also effective for untargeted clean-label backdoor attacks (e.g., UBW-C in [12])
>
> **A:** Thank you for your suggestion. We believe that UBW-C serves a different purpose, which is dataset ownership verification, whereas our study focuses on attacking the model. Consequently, we believe that explicitly providing the comparison with UBW-C may lead to potential confusion among the readers; additionally, we may also have to compare with additional dataset ownership verification methods for fair evaluation, which significantly enlarges the scope of our paper beyond clean-label backdoor attacks.
>
> We, however, believe that studying the applicability of our method in dataset ownership verification is an interesting extension of our work, which deserves an independent study. For example, as the algorithm in UBW-C does not select samples randomly but instead incorporates a selection strategy designed specifically for its optimization process, studying a better selection strategy for UBW-C, based on our work, is an interesting future direction.
>
> **Q9:** The Resistance to Defenses: The authors should evaluate their methods on more advanced backdoor defenses (such as [13, 14] and their baselines).
>
> **A:** We'd like to note that in our paper, we have already evaluated our strategy with **10** representative and recent backdoor defenses, ranging from backdoor erasing, training-data and inference-time backdoor detection, to anti-backdoor learning defenses; this makes our evaluation quite comprehensive in the backdoor domain with hundreds of backdoor defenses. Nevertheless, following the suggestion by the reviewer, we report the performance of our strategy against IBD PSC [13], a backdoor detection method. The results show that IBD PSC is not effective against low poisoning rate clean-label attacks, indicated by low AUC and F1 score; and our method does not make the attack less stealthy.
>
> | IBD PSC | Strategy   | AUC   | F1    |
> |---------|------------|-------|-------|
> | BadNets | Random     | 0.528 | 0.178 |
> |         | Pretrained | 0.549 |  0.240 |
> |         | OOD        |  0.550 | 0.279 |
> | Blended | Random     | 0.512 | 0.199 |
> |         | Pretrained | 0.502 | 0.152 |
> |         | OOD        | 0.518 | 0.189 |
> | SIG     | Random     | 0.516 | 0.116 |
> |         | Pretrained | 0.519 | 0.154 |
> |         | OOD        | 0.533 | 0.114 |

---

> > ### Comment · Reviewer_F14C · 2024-11-25
> >
> > I would like to thank the authors for providing their detailed feedback.
> >
> > 1. **Regarding the threat model**.
> > To the best of my knowledge, there are some works [1, 2] have discussed the setting of accessing samples only from the target class. Thus, although these methods were not designed under the task of sample selection, this threat model is not novel.
> >
> > 2. **Regarding the novelty**.
> > The technique of using OOD samples for optimizing the sample selection is similar to that used for optimizing trigger patterns used in [1, 2] to some extent, although under different settings.
> >
> > 3. **Regarding more advanced defenses**. As far as I know, the author only tested defenses that are 2 years old or even older in their original submission. As such, I think it is unfair to claim that 'We believe that our paper has compared our work with a wide range of backdoor attacks and defenses, from the most representative to the most recent ones.' in the rebuttal. However, I appreciate the efforts that the authors have made to add new experiments during the rebuttal.
> >
> > Regarding all previous aspects and my trade-off between the contributions of this paper, I decide to increase my score to 5.
> >
> > **References**
> > 1. Efficient backdoor attacks for deep neural networks in real-world scenarios
> > 2. Narcissus: A practical clean-label backdoor attack with limited information

---

> ### Author Response · Authors · 2024-11-25
>
> Thank you for your thoughtful comment and positive support. We'd like to further clarify our answers below.
>
> **Q1:** To the best of my knowledge, there are some works [1, 2] have discussed the setting of accessing samples only from the target class. Thus, although these methods were not designed under the task of sample selection, this threat model is not novel.
>
> **A:** We agree with the reviewer that our threat models share some resemblance to [1,2], as also discussed in our initial response; nevertheless, the true extent of the danger of these threat models were not fully exposed and in some cases, the attacks in these threat models are not even considered as serious.
>
> Specifically, one of the threat models studied in [1] is the class-constrained threat model, whose most restricted case is similar to our threat model. However, they observe that backdoor attacks in this threat model are impractical due to the low ASR or low stealthiness. In contrast, our work shows that smartly selecting suitable samples to poison significantly boosts the ASRs of several existing attacks (BadNets, Refool, etc...) while keeping the attacks' resilience against backdoor defenses; thus, our work is the first study that exposes the full extent of the vulnerability in this threat model.
>
> On the other hand, the threat model in Narcissus is similar to the second threat model in our work, which assumes that besides the target class's data, the adversary can have access to an OOD dataset. This threat model provides the attacker with more information, thus being more relaxed than the first threat model in our work where the attacker does not have any other data. Again, our work shows that by selecting data smartly for Narcissus, its ASR also improves significantly; in other words, our work again exposes an even more damaging attack based on Narcissus.
>
> In summary, our work has outlined complete scenarios for the threat model where the attacker only has access to the target class data, and has exposed the true danger of these scenarios when the attacker strategically selects the data to poison. We will include this discussion in the camera-ready version of the paper.
>
> **Q2:** The technique of using OOD samples for optimizing the sample selection is similar to that used for optimizing trigger patterns used in [1, 2] to some extent, although under different settings.
>
> **A:** We'd like to note that Narcissus [2] proposes a method to find the trigger pattern, whereas our sample selection strategy is an orthogonal approach and can be used in conjunction with different types of triggers, including Narcissus, as shown in Section B.6.
>
> **Q3:** As far as I know, the author only tested defenses that are 2 years old or even older in their original submission. As such, I think it is unfair to claim that 'We believe that our paper has compared our work with a wide range of backdoor attacks and defenses, from the most representative to the most recent ones.' in the rebuttal.
>
>
> **A:** In our paper, we have evaluated our strategy with recent backdoor defenses released last year, such as FT-SAM and RNP. Since our method preserves the utility of the attacks on **11** representative defenses, we can generally conclude that the expected performance of the attacks using our sampling strategy on other types of defense should be preserved. We will include the new results suggested by the reviewer in the camera-ready version of the paper.

---

### Official Review · Reviewer_PVcB · 2024-11-01

**Soundness:** 3
**Presentation:** 2
**Contribution:** 2
**Rating:** 6
**Confidence:** 4

**Summary:**

The paper proposes a method for improving the effectiveness of clean-label attacks. It introduces a threat model where the attacker only has access to data belonging to a specific class and has no knowledge about other classes in the dataset. The paper proposes a method for using samples with hard to learn features to create poison-efficient clean label attacks. The proposed method finds these samples by clustering the latent features of a surrogate model. The paper explores using a pretrained model and a model trained on OOD data as the surrogate model. The paper evaluates the clean-label attack against backdoor defenses and data cleaning methods.

**Strengths:**

- The proposed method is a widely applicable technique to enhance to clean label attacks.
- The experiments do a good job differentiating the surrogate model from victim model and therefore the attack shows convincing transferability.

**Weaknesses:**

- The paper trains for 300 epochs which is significantly longer than it should take to train the model on CIFAR-10/GSTRB [1,2] and makes attack success due to over-fitting very likely. Around 100 epochs seems to be more standard. Ideally, to simulate a competent defender early stopping should be employed. I.e. stopping the run when validation loss plateaus.

- The experiments use very weak baselines. The paper only evaluates how the method performs compared to random sampling. At minimum the paper should compare against [3]. Especially because [3] could easily be adapted to adhere to this paper's threat model by using a pretrained model. Therefore, the experiments are not sufficent to jusifty that the proposed method is stronger than a slightly adapted version of [3].

- The paper claims that it's threat model represents *"the **most** constrained data-poisoning threat."* However, there are other perfectly reasonable threat models that would make this attack unrealistic. For example, an opportunistic attacker that doesn't get to choose the subset of samples in the dataset they are able to manipulate.

- When evaluating the attack against defenses the paper does not describe the hyperparameter settings used by each defense nor how those settings were derived.

Minor:
- Bolding of best methods or aggregation would make Tables 2 and 3 more interpretable.
- There are many typos in the manuscript.

**Questions:**

- Why would an attacker use the OOD strategy proposed in section 4.4, as it requires training a surrogate model and appears to work worse than using a pretrained model?
- Why use a latent space clustering approach instead of using the loss from a pretrained zero-shot image classifier like CLIP?
- Why use VICReg instead of a more general feature extractor like CLIP?
- Where are the training settings used in experiments adapted from?

References

[1] Alexander Turner, Dimitris Tsipras and Aleksander Madry. "Label-Consistent Backdoor Attacks.", 2019.

[2] He, Kaiming, et al. "Deep Residual Learning for Image Recognition," 2016.

[3] Gao, Yinghua, et al. "Not All Samples Are Born Equal: Towards Effective Clean-Label Backdoor Attacks.", 2023.

---

> ### Author Response · Authors · 2024-11-23
>
> Thank you for your invaluable comments and for acknowledging our experiments. Please see our response below.
>
> **Q1:** The paper trains for 300 epochs which is significantly longer than it should take to train the model on CIFAR-10/GSTRB [1,2] and makes attack success due to over-fitting very likely. Around 100 epochs seems to be more standard. Ideally, to simulate a competent defender early stopping should be employed. I.e. stopping the run when validation loss plateaus.
> Where are the training settings used in experiments adapted from?
>
> **A:** We'd like to confirm that, for a fair evaluation, we follow the experimental setup used in the relevant works in the literature, such as Lin, et al. (300 epochs) and Zeng et al. (200 epochs), all of which train the model with more than 100 epochs. Some implementations, such as [Wanet](https://github.com/VinAIResearch/Warping-based_Backdoor_Attack-release/blob/main/config.py), even train the victim model with 1000 epochs. Nevertheless, we follow the reviewer's suggestion and conduct experiments with 100 epochs and report the results below; the experiments show that the number of epochs does not influence the effectiveness of our method.
>
> |                       | BadNets | Blended | SIG   |
> |-----------------------|---------|---------|-------|
> | Random                |   52.78 |   35.64 | 55.44 |
> | Self-supervised model |    69.50 |   55.51 | 78.79 |
>
>
> Lin, Tao, et al. "Don't Use Large Mini-batches, Use Local SGD." International Conference on Learning Representations. 2020.
>
> Zeng, Yi, et al. "Narcissus: A practical clean-label backdoor attack with limited information." Proceedings of the 2023 ACM SIGSAC Conference on Computer and Communications Security. 2023.
>
> **Q2:** The experiments use very weak baselines. The paper only evaluates how the method performs compared to random sampling. At minimum the paper should compare against [3]. Especially because [3] could easily be adapted to adhere to this paper's threat model by using a pretrained model. Therefore, the experiments are not sufficent to jusifty that the proposed method is stronger than a slightly adapted version of [3].
>
> **A:** [3] or Gao et al. (2023), cited and already discussed in our paper, propose three strategies to select samples by examining 1) loss value, 2) forgetting event, and 3) gradient norm, all of which cannot be adapted to launch the attack in our threat model (even when the pretrained model is used).  More particularly, computing the forgetting event requires monitoring the training process, while loss value and gradient norm can only be computed when the pretrained model can "return the target class". For example, the attacker cannot use a model pretrained on ImageNet to compute the loss value of images in the GTSRB dataset. Our strategy overcomes this challenge by detecting hard samples based on their features, or training a model on the target class and OOD data to compute the loss value.
>
> To the best of our knowledge, our sampling strategy is the only suitable method for the proposed threat model. Nevertheless, we are more than happy to promptly discuss and evaluate any missing and relevant work that the reviewer may recommend during the rebuttal process.
>
> Nevertheless, as suggested by the reviewer, we now report the performance of [3] when we relax the assumption of our threat model to allow access to other classes. More particularly, we train a clean model on CIFAR10 and compute the loss value to select hard samples. As can be observed, although having access to all training samples, the approach in Gao et al. does not outperform our strategy.
>
> |                   | BadNets | Blended | SIG   |
> |-------------------|---------|---------|-------|
> | Random            | 45.01   | 37.55   | 60.54 |
> | Gao et al. (2023) |   87.62 |   58.20 | 80.76 |
> | Ours (Pretrained) | 91.68   | 66.45   | 80.59 |
> | Ours (OOD)        | 81.27   | 56.89   | 80.76 |

---

> ### Author Response · Authors · 2024-11-23
>
> **Q3:** The paper claims that it's threat model represents "the most constrained data-poisoning threat." However, there are other perfectly reasonable threat models that would make this attack unrealistic. For example, an opportunistic attacker that doesn't get to choose the subset of samples in the dataset they are able to manipulate.
>
> **A:** Thank you for your comment. When the attacker cannot choose the subset of samples to manipulate, there are 2 cases: (1) the subset they can manipulate does not contain the target class samples, and (2) the subset contains some target class samples and several samples from other classes.
>
> The setting of (1) does not adhere to the conventional clean-label setting, which assumes access to target-class samples for manipulating the images but not changing the target label; consequently, we believe that it's extremely hard (if not impossible) to launch clean label attacks. For the setting of (2), we already provided such experiments in Section B.10, showing that when the attacker only has access to a subset of the target class our method is still effective.
>
> Note that, the "constraint" in backdoor attacks refers to the "information" the attacker can use to launch the attacks; while having less number of target class samples in (2) leads to less information on the target class, additionally having access to non-target class data also means "more information" from the other classes. Nevertheless, our strategy is still effective even if we only choose to use the information from a smaller set of target class data (as in the experiments in B.10).
>
> **Q4:** When evaluating the attack against defenses the paper does not describe the hyperparameter settings used by each defense nor how those settings were derived.
>
> **A:** Thank you for the comment. The results for all the defenses reported in our paper are obtained from the experimental settings in the corresponding original papers. We will add this statement in the camera-ready version of the paper.
>
> **Q5:** Why would an attacker use the OOD strategy proposed in section 4.4, as it requires training a surrogate model and appears to work worse than using a pretrained model?
>
> **A:** Thank you for your comment. The two data selection approaches cover two complementary scenarios for the attacker: when the pretrained model is available and when the pretrained model is not available (thus, one can be built from an OOD dataset). Choosing which attack approach consequently depends on this availability. When both options are available to the attacker, due to its observed higher performance in our experiments, the pretrained approach is a preferred choice, as mentioned in the comment. The second approach, i.e., OOD-strategy is indispensable when access to the pretrained model is unavailable. Here, the attacker can easily collect an OOD dataset; this attack is slightly less potent but still damaging, even when the OOD dataset is significantly distant as demonstrated in the paper.
>
> **Q6:** Why use a latent space clustering approach instead of using the loss from a pretrained zero-shot image classifier like CLIP?
> Why use VICReg instead of a more general feature extractor like CLIP?
>
> **A:** Thank you for your suggestion. The reason for not using multimodal pretrained models such as CLIP is that those models aim to align image features with textual features, thus, ignoring subtle visual details. For instance, CLIP tries to match an image of an English Springer with the prompt "a photo of a dog", therefore, CLIP is less likely to discriminate different dog species. This characteristic makes CLIP less effective in detecting outliers. To verify that, we conduct experiments where we select samples to poison by CLIP loss or kNN with CLIP features. The results show that this strategy is still better than the random baseline, however, it underperforms different methods where we use vision-only models to extract features. We will include this discussion in the camera-ready version of our paper to make the choice of pretrained models clearer.
>
> |                                  | BadNets | Blended | SIG   |
> |----------------------------------|---------|---------|-------|
> | Random                           | 45.01   | 37.55   | 60.54 |
> | CLIP (loss)                            | 87.75   | 50.90   | 71.80 |
> | CLIP (kNN)                            | 75.99   | 43.16   | 65.48 |
> | Self-supervised pretrained model | 91.68   | 52.90   | 80.59 |
> | Supervised pretrained model      | 92.14   | 60.86   | 85.42 |
>
> **Q7:** Bolding of best methods or aggregation would make Tables 2 and 3 more interpretable.
> There are many typos in the manuscript.
>
> **A:** Thank you for your comment. We will update them in the camera-ready version of the paper.

---

> > ### Comment · Reviewer_PVcB · 2024-11-24
> > **Thank you for your rebuttal.**
> >
> > First off, thank you for running so many additional experiments on my behalf.
> > 1. This experiment alleviates my overfitting concerns.
> > 2. My argument is that for a sufficently well trained CLIP model it can "return the target class" of basically any label by just comparing against the textual representation of that class. For example, the best trained CLIP checkpoints get >80% zero-shot accuracy on ImageNet-1k, see the github repo associated with [1]. Which is actually significantly higher than VicReg's reported accuracy of 73% on ImageNet. That is, even if it not a directly trained classifer for that dataset, it can still easily be used as one. Hence, you could use it to get loss values / grad norms and is therefore a reasonable baseline. I agree calculating the forgetting event is unrealistic.
> > 3. *the "constraint" in backdoor attacks refers to the "information" the attacker can use to launch the attacks*
> > Why would a "constraint" be information only? In my eyes a "constraint" in backdoor can be basically anything that constrains/limits attacker capabilities, and is therefore extremely broad. For example, [2] describes limiting accuracy degradation as a threat model "constraint".
> > 4. Thank you for including this.
> > 5. I think this loops back around to me seeing CLIP-like models as widely available generalist classifiers, so I would always assume a reasonable pretrained model is available. Overall, I think the authors perspective here is reasonable.
> > 6. Is this experiment using CLIP from OpenAI or OpenCLIP? What size/checkpoint?
> > 7. Good.
> >
> > **tl:dr** Overall the authors have done a good job addressing most of my concerns, I am happy to raise my score to a 5 if the authors revise *"this represents the most constrained data poisoning threat, wherein the attacker has extremely limited information for launching an effective attack."* and the similar statement in section 5.2 to clarify that they mean **information constrained**.
> >
> > [1] Reproducible scaling laws for contrastive language-image learning, Cherti et al.
> > [2] BadNets: Identifying Vulnerabilities in the Machine Learning Model Supply Chain, Gu et al.

---

> > > ### Author Response · Authors · 2024-11-24
> > > **Thank you for your insightful comment**
> > >
> > > We are grateful for your insightful comment and discussion.
> > >
> > > **Q2:** We completely agree with the reviewer that CLIP can be a reasonable baseline, which we provided its comparisons. Nevertheless, as explained, its performance is expected to be lower compared to the SSL models in our problem, even though CLIP has 80% zero-shot accuracy on ImageNet-1k, compared to
> > > the 73% accuracy of VicReg reported on ImageNet.
> > >
> > > In addition, zero-shot classification with CLIP is challenging in the case where the user wants to build a fine-grained classifier for their specific use case. For example, CLIP struggles to distinguish facial images of different people or images of different species of dogs, as discussed in the initial response of **Q6**. In contrast, vision-only pretrained models do a good job of detecting hard samples, as demonstrated in Section B.2 and experimental results in **Q6**. For the generality of the proposed method, the vision-only pretrained models, which capture general vision features, are more suitable.
> > >
> > > We will, however, include the CLIP baseline, and this discussion in the camera-ready version of our paper.
> > >
> > > **Q3:** We will clarify in the camera-ready version that the "constraint" in our paper means the information that the attacker has. Nevertheless, we also agree that the general sense of constraint is "anything that constrains/limits attacker capabilities"; in our work, we assume that all the baselines equally have these general constraints (e.g., limiting accuracy degradation), which should be satisfied.
> > >
> > > **Q5:** As responded in **Q2** above and in the initial response of **Q6**, we agree that CLIP-like models are reasonable for some settings. However, as selecting the right pretrained model with CLIP is generally less effective than our SSL approach (demonstrated in our experiments), we think that additional, independent research with CLIP could be an interesting extension of our work. In fact, we believe that this research direction with CLIP should study its suitability on poison data selection in other threat models as well.
> > >
> > > **Q6:** We use the ViT-B/32 CLIP model from the official implementation of [OpenAI](https://github.com/openai/CLIP).
> > >
> > >
> > > TL;DR: We will revise our paper accordingly (as mentioned above) in the camera-ready version. We hope that the reviewer will consider our discussions in the rebuttal to address the concerns, the importance of analyzing our threat model, and our contributions in the final rating!

---

> > > > ### Comment · Reviewer_PVcB · 2024-11-24
> > > >
> > > > Thank you for addressing my concerns. I have updated my rating accordingly.

---

> > > > > ### Author Response · Authors · 2024-11-25
> > > > > **Thank you for your support!**
> > > > >
> > > > > Thank you again for your constructive comments and suggestions. We're happy that our responses have addressed your concerns, and to receive your positive support on the paper.

---

### Official Review · Reviewer_BTAB · 2024-11-03

**Soundness:** 3
**Presentation:** 3
**Contribution:** 3
**Rating:** 8
**Confidence:** 5

**Summary:**

The paper studies clean-label backdoor attacks in a very constrained setting, where the attacker only needs access to the training data from the target class and has no prior knowledge of the victim model, training process, and the other classes, and focuses on data-selection strategies to boost the performance of existing clean-label attacks in this constrained setting. The proposed data selection strategies include (1) the use of a pretrained model (when such exists) or (2) the use of an OOD dataset (when the pretrained model is not available) to train a surrogate model.  The experimental results demonstrate the proposed strategies significantly enhance the ASR and several existing clean-label backdoor attacks, compared to random selection strategies. In addition, the paper demonstrates that the proposed strategies are resilient against several existing defenses.

**Strengths:**

The main strengths of the paper lie in its studied threat model, proposed sampling strategies and experimental evaluation.

- I think that the proposed threat model is important as it exposes yet another backdoor threat where an attack only needs access to the data of the target class. The demonstrations that existing backdoor attacks under this threat model are not satisfactorily effective are an important contribution of the paper.
- The proposed sampling strategies are novel, especially when they could be used with existing backdoor attacks, such as BadNets, SIG, Narcissus, etc…) to boost their backdoor performances under the studied threat model. It’s also interesting finding where the effectiveness of the proposed strategies even when there is less and less assumptions on the pretrained models or the OOD datasets.
- The paper includes thorough analysis of the proposed strategies, demonstrating their effectiveness when they’re used with several existing backdoor attacks across different datasets. The evaluation also shows the effectiveness against several defenses.

**Weaknesses:**

I find that the paper has the following concerns:
* The Narcissus results, while interesting, are different from what reported in their original paper. Can the authors explain why there are such differences?
* The OOD approach rely on out-of-distribution data but it’s not clear how this dataset could be obtained, or whether there are any specific requirements of the datasets to maintain the effectiveness of the attacks?
* Assuming that the victim could distribute the target class data collection to multiple sources, how does the proposed attacks perform in this case?
* Do the authors have any suggestions about potential mitigation approaches against the proposed attacks in the studied threat model?

**Questions:**

Please see the questions in weaknesses.

---

> ### Author Response · Authors · 2024-11-23
>
> We are grateful for your insightful comments, and for appreciating our threat model, analysis, and experiments and acknowledging the novelty of our strategies. Please see our response below.
>
> **Q1:** The Narcissus results, while interesting, are different from what reported in their original paper. Can the authors explain why there are such differences?
>
> **A:** We'd like to confirm that we used the official implementation of these attacks to poison the datasets. As indicated in Narcissus's paper and in their official repository (https://github.com/reds-lab/Narcissus/issues/3), Narcissus's results have a very high variance; consequently, the results in our paper are averaged over 3 random seeds under our experimental setting using their official implementation. As discussed in Section B.6, we conjecture that the high variance is due to sample selections; for easy samples, the ASR (13.06%) is significantly lower than that (56.16%) of random samples, which is significantly lower than the ASR (89.65%) with samples proposed by our method. This behavior further supports our analysis, showing the importance of selecting samples to poison.
>
>
> **Q2:** The OOD approach rely on out-of-distribution data but it’s not clear how this dataset could be obtained, or whether there are any specific requirements of the datasets to maintain the effectiveness of the attacks?
>
> **A:** As has been shown in the results on GTSRB in Table 3, and especially PubFig and ImageNet-Sketch in Section B.2, our OOD data approach can boost the attack success rate even when the OOD dataset is significantly far from the target dataset, for example, the attacker can employ a general dataset such as TinyImagenet while the victim is training the model on face recognition task. We conjecture that surrogate models pretrained on a general dataset can detect task-agnostic features, such as edges, shapes, colors, etc..., which are sufficient to find outliers in the target class. Therefore, the assumption of access to OOD data can be easily satisfied, showing the practicality of our method.
>
>
> **Q3:** Assuming that the victim could distribute the target class data collection to multiple sources, how does the proposed attacks perform in this case?
>
> **A:** Thank you for your comment. This scenario has already been studied in Section B.10 of our paper. Specifically, Tab. 20 shows that if the attacker only provides 20% of the target class, selecting hard samples in that set still increases the success rate by more than 20%.
>
> **Q4:** Do the authors have any suggestions about potential mitigation approaches against the proposed attacks in the studied threat model?
>
> **A:** Thank you for the interesting question. If the victim is aware of this type of attack, one defensive possibility is to utilize the same sample-selection strategy discussed in Section 4 to detect and remove outliers in the dataset. However, the attacker can also counter this adaptive defense by poisoning medium-hard samples; for example, this adversary can sort the training samples from hard to easy and choose those below the top 20%. Figure 2 demonstrates that this strategy is still an effective attack, which performs even better than the baseline.
>
> Currently, due to the important findings of our work, we encourage the model user to source dataset collection/annotation to trusted partners, to reduce the risks of the malicious actors using our methodology to launch a harmful backdoor attack. Otherwise, we believe that developing the countermeasure against our proposed attacks is non-trivial and deserves a separate, independent study as a future extension of our paper.

---

> > ### Comment · Reviewer_BTAB · 2024-11-23
> >
> > I want to thank the authors for their detailed response. All my concerns are well addressed. I believe this paper brings meaningful insights to the research community, thus I lean to accept the paper.

---

> > > ### Author Response · Authors · 2024-11-23
> > > **Thank you for your positive response!**
> > >
> > > Thank you again for your constructive comments. We're happy that our responses have addressed your concerns, and to receive your positive support on the paper.

---

### Official Review · Reviewer_8TyJ · 2024-11-04

**Soundness:** 2
**Presentation:** 3
**Contribution:** 2
**Rating:** 5
**Confidence:** 4

**Summary:**

The paper proposes a method for enhancing clean-label backdoor attacks on deep neural networks. Unlike traditional clean-label attacks that apply triggers randomly, this approach selectively poisons challenging samples within the target class, boosting attack success rates with fewer poisoned samples. The authors introduce two strategies: using pretrained models to identify "hard" samples and leveraging out-of-distribution data for sample selection. Tested on CIFAR-10 and GTSRB datasets, this method outperforms random poisoning and is resilient against popular defenses like STRIP and Neural Cleanse, highlighting a need for stronger countermeasures against selective clean-label attacks.

**Strengths:**

1. The paper improves traditional clean-label backdoor attacks by proposing a threat model that is more applicable in real-world scenarios.

2. The method is claimed to achieve higher attack success rates with a lower poisoning rate, showcasing efficient use of resources.

**Weaknesses:**

1. In my opinion, the method primarily introduces a data selection strategy, which lacks sufficient novelty.

2. The evaluation is conducted only on CIFAR-10 and GTSRB datasets, limiting insight into the method's performance across other dataset types and application domains.

3. The paper primarily tests against older defense strategies. Implementing more recent and sophisticated defenses, including adaptive methods like sample-specific anomaly detection, would strengthen the evaluation.

4. The pretrained model strategy relies on the availability of pretrained models in similar domains, which may not always be accessible in real-world applications.

**Questions:**

The authors should clarify the novelty, choice of limited datasets, the use of older defense strategies, and the dependency on pretrained models.

---

> ### Author Response · Authors · 2024-11-23
>
> We really appreciate your insightful feedback. Please see our response below.
>
> **Q1:** In my opinion, the method primarily introduces a data selection strategy, which lacks sufficient novelty.
>
> **A:** Thank you for the comment. As discussed in the paper and also discussed by Reviewers BTAB and F14C, the main novelty of the paper (and thus the contributions) includes the discovery of a new and extremely constrained threat model and the proposal of novel sample selection algorithms that can empower existing backdoor attacks to achieve harmful attack success rates. These contributions are significant in the backdoor domain, as they demonstrate the existence of a very stealthy and harmful attacker.
>
> Note that, existing works [Xia et al. (2022), Gao et al. (2023)] also study data selection but for the threat model where the attacker accesses data from all classes. Our work, on the other hand, demonstrates that even with limited knowledge (knowing data from one class) of the task, the attacker can still launch highly effective attacks using our novel sample-selection method. As discussed in Section 3, this threat model is popular in practice, where the victim struggles to or cannot collect the dataset and needs to rely on the third party.
>
> **Q2:** The evaluation is conducted only on CIFAR-10 and GTSRB datasets, limiting insight into the method's performance across other dataset types and application domains.
>
> **A:** In the paper, we have evaluated our approach with a wide range of datasets. More specifically, Sec. B.2 shows that even in the scenarios with extreme distribution shifts such as Imagenet-Sketch and PubFig, our method still increases the attack success rate, exposing a serious threat in practice.
>
> **Q3:** The paper primarily tests against older defense strategies. Implementing more recent and sophisticated defenses, including adaptive methods like sample-specific anomaly detection, would strengthen the evaluation.
>
> **A:** Thank you for your comment. We'd like to note that in our paper, we have already evaluated our strategy with **10** backdoor defenses, including recent defenses such as FT-SAM and RNP, ranging from backdoor erasing, training-data and inference-time backdoor detection, to anti-backdoor learning defenses. Experimental results show that our strategy is resilient to existing backdoor defenses while boosting the success rate significantly. Table 5 also indicates that current sample detection defenses are not effective against our method.
>
> On the other hand, the user can perform adaptive defense by utilizing the same sample-selection strategy discussed in Section 4 to detect and remove outliers in the dataset. However, the attacker can also counter this adaptive defense by poisoning medium-hard samples; for example, this adversary can sort the training samples from hard to easy and choose those below the top 20%. Figure 2 demonstrates that this strategy is still an effective attack, which performs even better than the baseline. We believe that developing the countermeasure against this type of attack is non-trivial and deserves a separate, independent study as a future extension of our paper.
>
> **Q4:** The pretrained model strategy relies on the availability of pretrained models in similar domains, which may not always be accessible in real-world applications.
>
> **A:** As discussed in the paper, when the pretrained models are not available, the attacker can employ an OOD dataset, with similar successes. We also show that, this strategy can still improve the success rate even when the distribution of the training dataset is significantly different from pretrained models (with GTSRB, PubFig, and Imagenet-Sketch). The attacker can employ general pretrained models to detect hard samples in the target class to poison. Therefore, the assumption of pretrained models can be easily satisfied, showing the practicality of our method.

---

> > ### Comment · Reviewer_8TyJ · 2024-11-26
> > **Thanks for the response**
> >
> > I appreciate the authors' effort to address the raised points and provide additional context. However, I have several concerns (related to my previous concerns) that I believe warrant further discussion and clarification:
> >
> > 1. Threat Model Novelty: While the authors describe their approach as addressing a "new and extremely constrained threat model," I respectfully disagree. Many existing works on label-free attacks (e.g., Refool, WaNet, Learnable Imperceptible Robust Backdoor Attack) have already demonstrated scenarios or through minor modifications that can make triggers be injected solely into the target class, thus negating the need for knowledge of all classes. These prior works suggest that such a "single-class knowledge" assumption is not unprecedented. Could the authors elaborate further on how their threat model significantly differs from these works in terms of knowledge limitation?
> >
> > 2. Dataset Selection: The datasets used, particularly PubFig, are relatively small in the context of backdoor attacks. As mentioned, the authors further reduced PubFig to around 5000 samples for training, which makes it even less representative of real-world challenges. In contrast, larger datasets such as ImageNet-1K or TinyImageNet are commonly used in backdoor attack and defense papers (e.g., Universal Backdoor Attacks, How to Inject Backdoors with Better Consistency). These datasets better capture the complexity of practical scenarios and are also leveraged in backdoor defense evaluations. It seems to me that the TinyImageNet dataset in this paper is only leveraged for a pre-trained model.
> >
> > 3. Defense Evaluation: There are many powerful and relevant recent defenses that have not been considered, such as: SCALE-UP: An efficient blackbox input-level backdoor detection via analyzing scaled prediction consistency; Distilling cognitive backdoor patterns within an image; ASSET: Robust backdoor data detection across a multiplicity of deep learning paradigms; How to sift out a clean data subset in the presence of data poisoning, etc.

---

> > > ### Author Response · Authors · 2024-11-30
> > >
> > > Thank you for your comment. Please see our response to the remaining concerns below.
> > >
> > > **Q1:** Threat Model Novelty.
> > >
> > > **A:** We agree with the reviewer that there are backdoor attacks that create the trigger without the information of the dataset. However, as already discussed in the paper, these attacks are either not stealthy or not effective in the clean-label setting. For example, we perform clean-label backdoor attacks with Wanet on CIFAR10 and achieve 11.91% ASR even with 100% poisoning rate on the target class.
> > > Label-Consistency, Refool, and Narcissus have shown the challenges of clean-label attacks and proposed to use additional information to create the trigger or make the attack stronger. More specifically, Label-Consistency uses an adversarially trained model to make input harder to learn before injecting the trigger; Refool suggests an intensive training loop to select adversarial reflection images; and Narcissus requires training a surrogate model to create the trigger pattern. We also would like to note that LIRA requires bilevel optimization on the training set to find the trigger and is not a clean-label attack.
> > >
> > > On the other hand, our approach significantly boosts the success rate of clean-label attacks while maintaining stealthiness with the information of the target class only. Furthermore, our method can be combined with different types of triggers and shows convincing transferability, as mentioned by Reviewer PVcB.
> > > Our work exposes the full extent of the vulnerability in this threat model, thus being practical and significant, as acknowledged by Reviewer BTAB and in the initial review.
> > >
> > > **Q2:** Dataset Selection.
> > >
> > > **A:** We completely agree with the reviewer that it's important to conduct experiments on large datasets that capture the complexity of practical scenarios. Thus, we have already reported the performance of our strategy on TinyImagenet in Section B.5 of the original manuscript. Experimental results show that our method is still effective on large datasets, indicating the practicality of our approach. In addition, our paper has provided extensive evaluations across multiple datasets, more than those evaluated in many related and representative backdoor papers such as BadNets, Refool, WaNet, LIRA, or Narcissus.
> > >
> > > **Q3:** Defense Evaluation.
> > >
> > > **A:** In our paper, we choose to experiment with many representative backdoor defenses, making our evaluation quite comprehensive in the backdoor domain with hundreds of backdoor defenses. Nevertheless, following the suggestion by the reviewer, we conduct experiments with other backdoor defenses, which are SCALE-UP, Cognitive Distillation, and ASSET. We report the AUC and F1 score of SCALE-UP, AUROC of Cognitive Distillation on the training set and test set, and true positive rate (TPR) and false positive rate (FPR) of ASSET below, showing that our strategy does not make the "base" attack significantly less stealthy under these defenses.
> > >
> > > Note that, as consistently discussed in our paper, our proposed sampling algorithms do not reduce the stealthiness of the "base" attack against a defense. As a result, in the future, if a powerful defense is overcome by a new attack, the adversary can use this new attack with our sampling strategy (in our threat model) to make this new attack suitable for our threat model while achieving stealthiness against the powerful defense. This shows the significant generality of our proposed sampling strategies.
> > >
> > > | SCALE-UP | Strategy   | AUC   | F1    |
> > > |----------|------------|-------|-------|
> > > | BadNets  | Random     | 0.611 | 0.511 |
> > > |          | Pretrained | 0.487 | 0.441 |
> > > | Blended  | Random     | 0.742 | 0.543 |
> > > |          | Pretrained | 0.612 | 0.565 |
> > > | SIG      | Random     | 0.468 | 0.371 |
> > > |          | Pretrained | 0.464 | 0.371 |
> > >
> > > | CD         | BadNet      | Blended     | SIG         |
> > > |------------|-------------|-------------|-------------|
> > > | Random     | 0.738/0.527 | 0.504/0.558 | 0.941/0.712 |
> > > | Pretrained | 0.763/0.569 | 0.662/0.527 | 0.803/0.687 |
> > >
> > >  | ASSET   | Strategy   | TPR   | FPR   |
> > > |---------|------------|-------|-------|
> > > | Badnets | Random     | 0.708 | 0.543 |
> > > |         | Pretrained | 0.624 | 0.534 |
> > > | Blended | Random     | 0.332 | 0.561 |
> > > |         | Pretrained | 0.572 | 0.542 |
> > > | SIG     | Random     | 0.697 | 0.615 |
> > > |         | Pretrained | 0.328 |  0.540 |

---

> ### Comment · Reviewer_8TyJ · 2024-12-03
> **Follow-Up on Response**
>
> Thank you for your detailed responses. I appreciate the inclusion of more defenses in your evaluation, which provides a broader understanding of your method's performance. However, I strongly suggest conducting further experiments under varied settings, such as different poisoning ratios, architectures, and other factors, to provide a more comprehensive analysis of the attack against those defenses. Regarding the novelty of the attack and the threat model, I remain unconvinced. While I acknowledge your efforts to highlight their significance, I find the arguments presented insufficient to alter my current perspective. Therefore, I will maintain my score.

---

### Comment · Area_Chair_yZ7e · 2024-11-24

Dear reviewers,

Thanks for serving as a reviewer. As the discussion period comes to a close and the authors have submitted their rebuttals, I kindly ask you to take a moment to review them and provide any final comments.

If you have already updated your comments, please disregard this message.

Thank you once again for your dedication to the OpenReview process.

Best,

Area Chair

---

### Author Response · Authors · 2024-12-04
**Summary of our responses and paper revision during the rebuttal phase**

We'd like to thank the reviewers for their constructive and helpful comments during the rebuttal and discussion phases. During these phases, we have:

- Clarified the novelty of our threat model and sample selection strategy. Our threat model is important and realistic, as acknowledged by Reviewer BTAB and F14C. Reviewer BTAB and PVcB appreciate the novelty and transferability of our strategy.
- Provided additional results of 4 backdoor defenses as suggested by Reviewer 8TyJ and F14C, proving that our method is robust against a wide range of backdoor defenses as we had shown in the original manuscript (10 defenses already provided in the submitted version).
- Clarified that our approach does not require the pretrained model or the OOD dataset to have the same domain as the victim dataset, showing the practicality of our method.
- Discussed a scenario where the victim could distribute the target class data collection to multiple sources in response to Reviewer BTAB. Our strategy is still threatening in this case as shown in the original manuscript.
- Clarified the experimental setting in our paper.
- Provided additional results of the strategy in Gao et al. Although having less information, our strategy is not less effective than that method.
- Provided additional results of using CLIP to select samples and discussed why CLIP does not perform as well as other vision-only pretrained models chosen in this work.
- Clarified the constraint setting in our threat model. We propose the most constrained threat model in terms of the amount of information the attacker can use to launch the clean-label attacks.

We have added these discussions in the revised submissions. We would be grateful if the reviewer could kindly consider our responses in their opinions when making the final decision, as we believe that we already addressed all their concerns. Again, thank you for the helpful and valuable comments.

---

### Public Comment · ~Quang_H_Nguyen2 · 2025-04-13
**Camera-ready update**

We'd like to update the results of our strategy against IBD-PSC defense due to a minor bug in the previous implementation. The table below demonstrates that although IBD-PSC can detect BadNets, our method with Blended and SIG trigger is still stealthy under this defense, showing the significant threat of our method.



|  | Strategy   | AUC   | F1    |
|---------|------------|-------|-------|
| BadNets | Random     | 0.999 | 0.955 |
|         | Pretrained | 0.999 | 0.976 |
|         | OOD        | 0.995 | 0.950 |
| Blended | Random     | 0.861 | 0.331 |
|         | Pretrained | 0.523 | 0.000     |
|         | OOD        | 0.848 | 0.321 |
| SIG     | Random     | 0.788 | 0.238 |
|         | Pretrained | 0.728 | 0.049 |
|         | OOD        | 0.869 | 0.273 |

We've updated this result in the camera-ready version of the paper.

---

### Meta-Review · Area_Chair_yZ7e · 2024-12-21

**Metareview:**

This paper examines clean-label backdoor attacks in a highly constrained setting, where the attacker only has access to training data from the target class and lacks prior knowledge of the victim model, training process, or other classes. In this senario, the authors manage to propose a data-selection strategy to improve the performance of clean-label attacks, like using a pre-trained model or OOD trained surrogate model. Experimental results demonstrate the methods effectiveness.

Strength:

1. The paper introduces a clean-label backdoor attack that works effectively in a constrained scenario where the attacker has limited data access (only one target class). This approach is realistic for scenarios with privacy or geographical constraints, enhancing the practical relevance of the attack model.

2. Comprehensive experiments to demonstrate their effectiveness.

Weakness:

Evaluation settings are not enough. For example, they only test ResNet-18 and VGG. If transformers are also involved, the impact will be greater.

Most reviewers show postive attitude towards this paper. The only remaining conerns are still need to adding empirical settings. However, regarding that the paper has already done a lot evaluations. I think missing some settings are acceptable. Therefore, I tend to accept this paper.

**Additional Comments On Reviewer Discussion:**

The reviewers and authors discussed about the empirical settings, novelty and etc. Although some authors still worried about lacking some baselines or empicial setting, I think the current results are enough to demonstrate the method's effectiveness considering the basline attacks, defenses are too many to include all of them and the current paper has already include a lot.

---

### Decision · Program_Chairs · 2025-01-22

Accept (Poster)